# A constricted opening in Kir channels does not impede potassium conduction

Katrina A. Black[1,2,9], Sitong He[3,9], Ruitao Jin [3,9], David M. Miller[1,2,9], Jani R. Bolla [4], Oliver B. Clarke[5], Paul Johnson[6], Monique Windley[7], Christopher J. Burns [8,2], Adam P. Hill[7], Derek Laver[6], Carol V. Robinson [4], Brian J. Smith [3✉] & Jacqueline M. Gulbis [1,2✉]

The canonical mechanistic model explaining potassium channel gating is of a conformational change that alternately dilates and constricts a collar-like intracellular entrance to the pore. It is based on the premise that K$^+$ ions maintain a complete hydration shell while passing between the transmembrane cavity and cytosol, which must be accommodated. To put the canonical model to the test, we locked the conformation of a Kir K$^+$ channel to prevent widening of the narrow collar. Unexpectedly, conduction was unimpaired in the locked channels. In parallel, we employed all-atom molecular dynamics to simulate K$^+$ ions moving along the conduction pathway between the lower cavity and cytosol. During simulations, the constriction did not significantly widen. Instead, transient loss of some water molecules facilitated K$^+$ permeation through the collar. The low free energy barrier to partial dehydration in the absence of conformational change indicates Kir channels are not gated by the canonical mechanism.

[1] Structural Biology Division, The Walter and Eliza Hall Institute of Medical Research, Parkville, VIC 3052, Australia. [2] Department of Medical Biology, The University of Melbourne, Parkville, VIC 3052, Australia. [3] La Trobe Institute for Molecular Science, La Trobe University, Melbourne, VIC 3086, Australia. [4] Physical and Theoretical Chemistry Laboratory, University of Oxford, Oxford OX1 3QZ, UK. [5] Biochemistry and Molecular Biophysics, Columbia University, New York, NY 10032, USA. [6] School of Biomedical Sciences and Pharmacy, The University of Newcastle, Newcastle, NSW 2300, Australia. [7] Computational Cardiology Laboratory, Victor Chang Cardiac Research Institute, Darlinghurst, NSW 2010, Australia. [8] Chemical Biology Division, The Walter and Eliza Hall Institute of Medical Research, Parkville, VIC 3052, Australia. [9] These authors contributed equally: Katrina A. Black, Sitong He, Ruitao Jin, David M. Miller. ✉email: brian.smith@latrobe.edu.au; jgulbis@wehi.edu.au

Potassium channels propagate electrical signals, maintain membrane potentials and facilitate numerous downstream cellular processes. To control ion permeation, K[+] channels switch between activated, resting and unresponsive (inactivated) states. The reversible transition distinguishing the conducting and non-conducting states of the pore has been explained by a conformational change causing an intracellular opening to the conduction pathway to expand sufficiently to accommodate fully hydrated K[+] ions. The inward rectifier K[+] (Kir) channels are important in brain and cardiac function[1–12] and are notable for their ability to limit K[+] efflux from cells[13,14]. Upon membrane depolarization, linear polyamines (e.g. spermine) released from a shallow cation-binding site in the cytoplasmic assembly of the channel penetrate deeply into the pore and stem, or rectify, outward K[+] flow[15,16] in a process known as open-channel block. Structures of Kir potassium channels superimpose over the entire transmembrane pore, whether prokaryotic[17,18], eukaryotic[19,20] or chimeric[21]. All feature an hourglass-like constriction at the intracellular opening, leading to the widespread conclusion that the canonical permeation gate at the helix bundle crossing is closed (or nearly so) in all Kir structures, whether determined by crystallography or single-particle cryo-EM. The opening is not wide enough to accept a fully hydrated K[+] ion, but is sufficient to accept an extended linear polyamine[18]. The pore conformation is not significantly influenced by crystal packing forces; in almost all Kir structures, the lattice contacts between adjacent proteins are mediated by the intracellular domains, while the transmembrane domains are solubilised in detergent micelles and make minimal interactions with neighboring molecules.

Structures of Kir channels in association with potentiating ligands do not exhibit the increase in the width of the conduction pathway in a conducting channel predicted by the canonical model. In crystal structures of Kir2.2 with the activating ligand phosphatidylinositol-4,5-bisphosphate (PI(4,5)P$_2$) bound[20], the pore remains as narrow as in the apo-structure[20]. The Kir3.2 (GIRK2, the KCNJ6 gene product) channel co-crystallized with an activating Gβγ protein complex[22–24], has an intracellular opening only incrementally wider than other Kir channels[25]. ATP and ADP-bound structures of K$_{ATP}$ (a Kir6.2/ sulfonylurea–receptor complex) determined by single-particle cryo-EM reveal equally narrow pore conformations[26]. Moreover, although single S129R[27] and double S129R/S205L[28] mutants of KirBac3.1 show subtle divergence of the inner helix backbone, the intracellular opening, delineated by a 6 Å distance between nitrogen atoms of opposing arginine guanidiniums within the conduction pathway, is no wider than wild type.

The lack of certainty circumscribing gating mechanism in Kir channels prompted us to investigate the requirement for, and degree of, conformational change accompanying gating and conduction. We carried out molecular dynamics to evaluate the free-energy cost incurred by ions passing from the pore cavity through a constriction to the intracellular solution. In parallel, we tested whether KirBac3.1 channels, with their inner helices constrained together by covalent linkages encircling the intracellular mouth, retain the ability to function. Inter-subunit disulfide bonds and chemical cysteine-crosslinkers were employed to afford some latitude in the tightness of crosslinking. We rationalized that loss of function would only result if conduction was contingent on pore widening, and that the use of different crosslinkers or pore blockers might be useful to gauge the approximate opening size required for K[+] to pass. In parallel, we employed all-atom molecular dynamics to determine the free-energy barrier that would be faced by K[+] passing through a tight constriction. Crosslinking the pores did not prevent K[+] conduction but did prevent entry of a narrow blocker of Kir channels, spermine, which has a cross-sectional width between that of the

ionic and hydrated diameters of K[+]. Thus, data presented here indicate that inward rectifiers (Kir) are able to conduct without requiring the substantial conformational changes predicted by the canonical gating mechanism. Instead, each K[+] transiently loses water from its hydration shell as it passes through. On the basis of these findings, we conclude that the fundamental mechanism of controlling currents in Kir channels, in some cases triggered by partner proteins, is presently unknown.

## Results

**A 2.0 Å crystal structure delineates the conduction pathway.** The molecular architecture of Kir channels has been described previously[17,18]. A near atomic resolution crystal structure (2.0 Å) of wild-type KirBac3.1 (Supplementary Table 1 and Supplementary Fig. 1) was determined, providing an accurate starting model for simulations. KirBac3.1 adopts the canonical fold of a K[+] channel, with inner and outer membrane-spanning helices supporting an inclined, shorter, pore helix that supports the signature selectivity filter (Fig. 1a). Although the internal cavity is not particularly narrow, it rapidly tapers at either end, delineated by Tyr132 and Thr96 (Fig. 1b, c).

**MD simulations address free-energy barriers to permeation.** To determine the energetic barrier that would be encountered by K[+] ions passing through the constriction located at Tyr132 in the inner helix bundle crossing, we adapted a molecular dynamics (MD) method previously employed to determine the free-energy landscape for gramicidin A[29]. Steered molecular dynamics simulations, in which K[+] ions are moved from the transmembrane cavity to the cytosol, were carried out on the pore of KirBac3.1 (residues 33–138 of the 2.0 Å structure) embedded in a POPC membrane separating saline reservoirs. Initially, K[+] ions were included in the selectivity filter and internal cavity (upper and lower sites), along with a surfeit of water molecules in the cavity. In addition, a homology model was prepared in which cysteine substitutions at residues 129 and 135 formed cystine bridges between adjacent inner helices, encircling the conduction path at Tyr132; after minimization, this model was subject to simulation protocols, as for wild type. Umbrella sampling simulations (Supplementary Fig. 2 and Supplementary Table 2) were performed on structures extracted from a 1-dimensional pulling trajectory along the molecular axis, and a weighted histogram analysis method (WHAM)[30] used to calculate the potential of mean force (PMF) experienced by each ion. Detailed methods are provided in the Supplementary Information.

Over the duration of the simulations the narrowness of the shallow intracellular opening did not prevent K[+] passing through. The simulations report free-energy barriers at the Tyr132 collar of 6 kJ mol[−1] for wild type and 1 kJ mol[−1] for the disulfide-linked mutant. These values fall within the range expected for thermal fluctuations and are too low to impair K[+] flux (Fig. 2a, b).

**The hydration shell of K[+] ions is transiently depleted.** Physical factors that determine the size of aperture through which an ion can pass are its coordination number, which defines the molecular volume, and its propensity to release and exchange ligands. In the selectivity filter, each K[+] rapidly exchanges between four eight-coordinate binding sites, whereas in solution a predicted coordination number of six to seven[31,32] is consistent with observation. Notably, theoretical ab initio quantum mechanical calculations concur with the findings of diffraction and spectroscopic studies in coming to the conclusion that in solution K[+] is coordinated by two subpopulations of water: four tightly and two loosely bound water molecules[33].

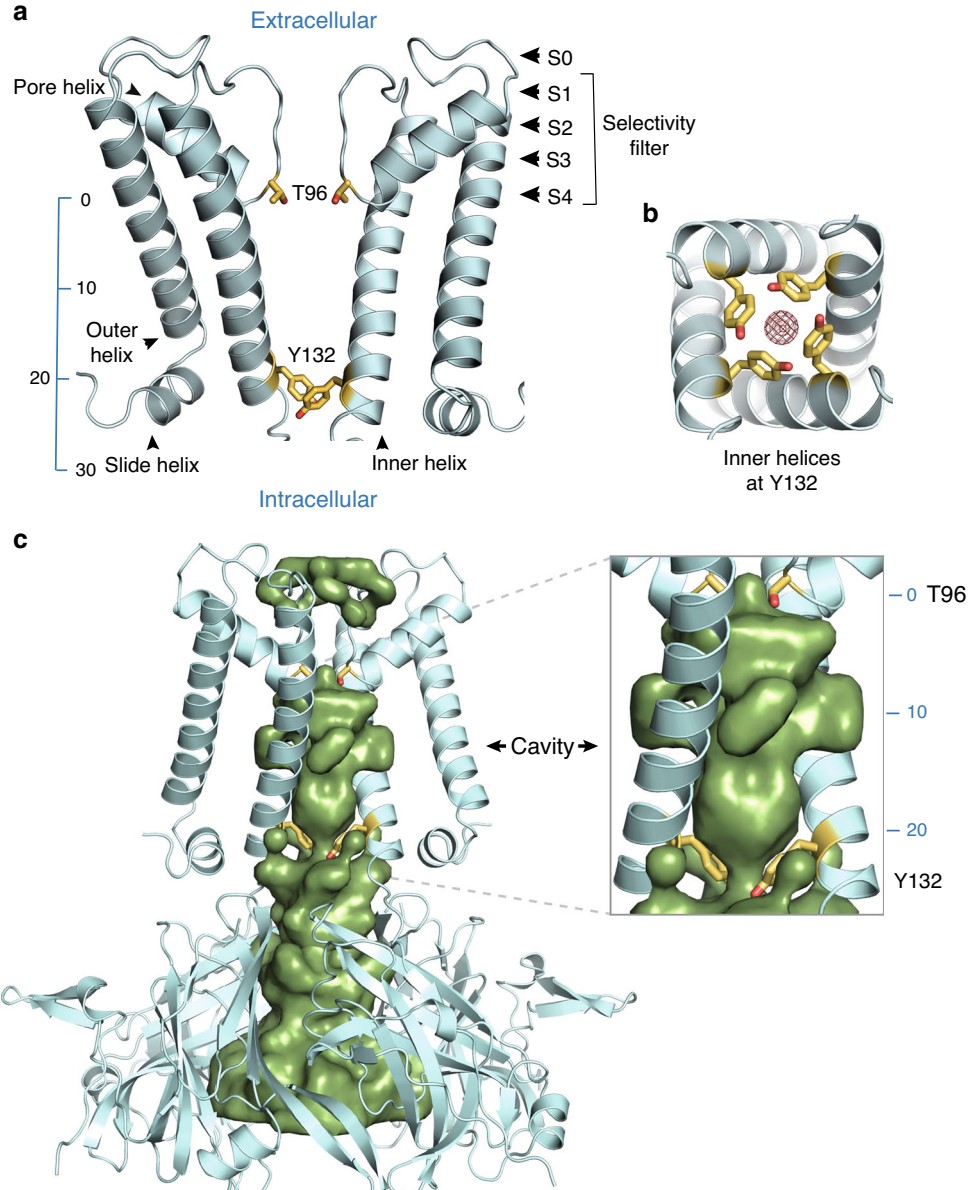

**Fig. 1 Features of the KirBac3.1 pore. a** Ribbon diagram of the wild-type KirBac3.1 pore in which two subunits are removed for clarity. A scale bar is provided indicating the distance (Å) from the center of mass of Thr96 and the potassium-binding sites S0 to S4 in the selectivity filter are indicated. **b** Transverse section showing that the conformation of the four inner helices maintains a narrow constriction at Tyr132. Density corresponding to $K^+$ ions is shown as brown mesh. **c** Accessible pore surface (green) along the molecular tetrad of wild-type KirBac3.1. Surface calculations were performed with HOLLOW. Two pore domain subunits have been removed for clarity. On the right is a close-up of the cavity through the membrane, showing how the cavity tapers to an hourglass at Tyr132. Tyr132 and Thr96 are represented as sticks throughout.

To evaluate the hydration state of $K^+$ during conduction, coordination numbers were determined from sampling data as a function of linear distance from the selectivity filter along the conduction pathway by enumerating the number of oxygen atoms within 3.0 Å of each $K^+$. Within the pore cavity, $K^+$ diffuses with a coordination number of approximately six. A fleeting reduction in the number of solvating water molecules to three or four occurs as the ion passes through the collar of tyrosine side chains but is almost immediately replenished. Transient interactions between $K^+$ and the Tyr132-hydroxyl groups occur after the ions pass the low energetic barrier near the center of the tyrosine aromatic ring. The total coordination number at each point along the $z$ axis, including the Tyr-OH

interactions, is presented (Fig. 2c, d and Supplementary Fig. 3). Potassium ions exhibit a slightly higher coordination number in wild type than in the disulfide-linked S129C-A135C mutant, but the mutant shows a small relative gain in hydroxyl coordination; this may be due to geometric constraints imposed on the Tyr132 side chain by the disulfide links. The retention of a kernel of water molecules means the free-energy barrier to permeation at the constriction is not prohibitive (Fig. 2e–g).

To detect any relationship between $K^+$ penetration and the size and shape of the aperture at the Tyr132 collar, small fluctuations in the internuclear distances between diagonally opposed Tyr132-hydroxyl oxygens were statistically analyzed (Supplementary Fig. 4). All 11 structures extracted at the precise instant individual

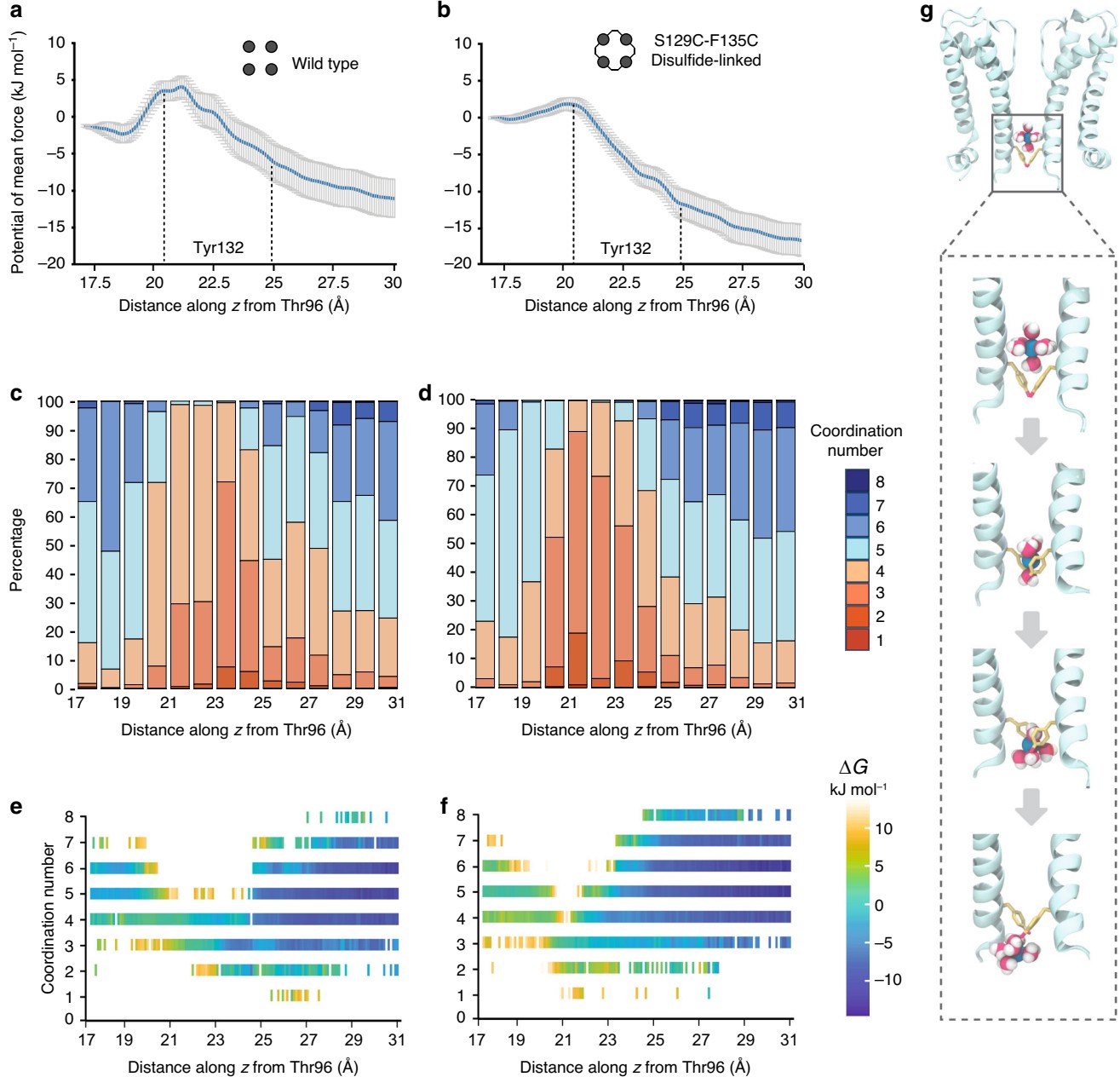

**Fig. 2 Molecular dynamics simulations indicate that K$^+$ passes through the Tyr132 collar in a partially hydrated state.** Wild-type panels are shown in **a**, **c** and **e**, and a disulfide-linked mutant in **b**, **d** and **f**. **a** The potential of mean force along the axis of the wild-type structure is shown as a function of the distance between a K$^+$ cavity ion 'pulled' along the conduction pathway (z axis) and the center of mass of the four Thr96 side chains (the global reaction coordinate). SEM are depicted as vertical gray bars and represents n = 200 bootstraps. The Tyr132 side chain position from Cβ (~20 Å) to OH (~25 Å) is indicated by dotted lines. **b** A comparable plot for disulfide-linked S129C-F135C. **c** Normalized histograms enumerate the number of oxygen atoms coordinating K$^+$ as a function of distance from Thr96. Data are accumulated from umbrella sampling simulations of wild-type KirBac3.1. Each bin represents the number of oxygen atoms within 3.0 Å of K$^+$, expressed as a percentage. **d** Histograms for disulfide-linked S129C-F135C. **e** Potential of mean force (PMF) for the wild-type channel is depicted as a function of the distance along the molecular axis from Thr96 and the number of coordinating ligands per ion. **f** PMF for disulfide-linked S129C-F135C. Energy units are kJ mol$^{-1}$. **g** Selected structures from simulations revealing K$^+$ ions in a partially dehydrated state as they pass Tyr132. Potassium ions are represented as green spheres and water depicted in red and white.

K$^+$ ions pass through the Tyr132 collar were compared to the entirety of the data (Supplementary Fig. 4). In all cases, the ions permeated mildly asymmetric Tyr132 collars with openings inadequate to accommodate fully hydrated ions (Supplementary Fig. 5). The mean values for the shorter and longer Tyr132-OH diagonals of the 11 structures correspond to van der Waals (vdW) spacings of 4.0 and 6.0 Å. (Supplementary Fig. 6), smaller than the time-averaged hydration diameter of K$^+$ of 6.6 Å[34].

**The conformation of the pore is constrained by crosslinking.** To covalently constrain the dimensions of the hourglass-like constriction, paired cysteine point mutants were introduced into KirBac3.1 upon a cysteine-less background, in which each of the three native cysteine residues were substituted for serine or valine (C262S, C71V, C119V). This allowed us to covalently link the four inner helices into a tetramer without obstructing the conduction pathway. Two mutants (A133C-T136C and S129C-

F135C) were constructed and utilized for structure determination and functional experiments (Fig. 3a).

Following purification of A133C-T136C and S129C-F135C, covalent linkages between adjacent subunits were generated, either by direct disulfide formation using $Cu^{2+}$ as the oxidant or by the crosslinking reagents 1,1-methanediyl bismethanethiosulfonate (MTS-1-MTS), which forms compact linkages ($^{Cys}S-S-CH_2-S-S^{Cys}$) with slightly more tolerance than direct disulfides ($^{Cys}S-S^{Cys}$), and dibromobimane, a rigid bicyclic molecule. In all, six unique mutant-crosslinker combinations were constructed.

Crosslink formation was validated by denaturing polyacrylamide gel electrophoresis, native mass spectrometry and X-ray crystallography. A change in migration on SDS-PAGE signified formation of covalent links (Fig. 3b, c). Native mass spectrometry demonstrated that disulfide-bonded and crosslinked tetramers fail to dissociate in the gas phase under conditions that release wild-type subunits (Fig. 3d). Recorded masses were consistent with tetrameric species with four crosslinks in each (Supplementary Table 3). Mass peaks were not observed for monomeric, dimeric or trimeric species, while incomplete crosslinking was excluded by the difference between calculated and observed masses. Native mass spectrometry verified the presence of covalent tetramers in all six variants.

To visually inspect the channels for intact crosslinks or unanticipated effects on structure, crystal structures of two of the KirBac3.1 adducts, S129C-F135C-MTS-1-MTS and A133C-T136C-bimane (Supplementary Table 1) were determined. In both mutants, four crosslinks encircle the conduction pathway at the Tyr132 as anticipated (Fig. 3e). Both revealed electron density bridging between adjacent inner helices, consistent with the predicted covalent linkages (Fig. 3f). An overlay of both crosslinked structures onto the native pore is illustrated (Fig. 3g).

**Crosslinked Kir mutants conduct $K^+$ currents**. KirBac3.1 and fully crosslinked mutants were reconstituted into liposomes and a well-established fluorimetric liposomal flux assay was used to measure activity. ACMA-based assays are essentially qualitative, but are adaptable to different ions and have been used to verify function in many types of ion channel, including a voltage-dependent proton channel $H_V1$[35], the Kir channel Girk2[36], a K2P channel K2P-TRAAK[37], the store-operated calcium channel Orai-1[38], a monovalent cation channel TRIC[39], and a chloride-selective channel CLC-K[40]. In this assay, a decrease in fluorescence due to protonation of the fluorophore 9-amino-6-chloro-2-methoxyacridine (ACMA) indicates $K^+$ permeation through the reconstituted channels (Fig. 4a). The fluorescence signal is normalized to the values immediately after addition of the protonophore carbonyl cyanide m-chlorophenylhydrazone (CCCP) and after addition of the $K^+$-specific ionophore valinomycin.

Examples of normalized data (Fig. 4b) and a data summary are presented (Fig. 4c and Supplementary Table 4) (Supplementary Fig. 7). In both wild-type and cysteine-less KirBac3.1 proteoliposomes, the fluorescence decrease occurring post CCCP addition indicates $K^+$ flux occurs through the $K^+$ channels. Control samples comprising liposomes reconstituted with buffer-only were used to determine baseline fluorescence. A 'null' liposome reconstitution control, prepared from comparably purified material obtained by isopropyl-β-D-thiogalactopyranoside (IPTG) induction of E. coli cells transformed with an empty expression plasmid, showed baseline signal. Moreover, tryptic digest mass spectrometry of samples prior to reconstitution showed no detectable contamination by E. coli ion channels, transporters or porins. To verify selectivity for $K^+$, the salts in the inner and outer buffer solutions were respectively altered from

$K^+_i/Na^+_o$ to $Na^+_i/N$-methyl-D-glucamine $(NMDG^+)_o$ (i = inside buffer; o = outside buffer) and a less specific ionophore, monensin, substituted for valinomycin. Baseline fluorescence (i.e. the same as empty liposomes) was observed in the $Na^+_i/NMDG^+_o$ experiment on wild-type KirBac3.1, consistent with a $K^+$ selective pore. To validate that the $K^+$ conductance arises specifically from Kir $K^+$ channels, a classical Kir channel blocker (spermine) was added to inner and outer solutions of proteoliposomes reconstituted with wild-type KirBac3.1; this abolished signal, indicating the $K^+$ flux occurred via KirBac3.1.

The cysteine-pair mutants were assayed under identical experimental conditions to wild type. All eight variants of the S129C-F135C or A133C-T136C mutants, either with reduced cysteine sulfhydryl moieties or as covalently-linked adducts with inner helices tethered at Tyr132, mediated $K^+$ movement out of liposomes, with no significant difference in signal from wild type in any of the crosslinked populations (Fig. 4c), indicating that crosslinking did not prevent $K^+$ flux through the Kir channels. The differences in rate (and overall signal) between crosslinked and reduced forms of the pore are not significant, which indicates the enforced constriction is not rate limiting. The rates ($s^{-1}$) and experimental time constants (tau) have been included as Supplementary Table 5.

Although ACMA assays do not replicate the fast physiological rates of ion channel function, published titration studies demonstrate their internal consistency[41]. The rates calculated from our experimental curves of KirBac3.1 fall in the mid-range of rates reported for Kir3.2 titrations under very similar conditions[41,42], indicating that in the context of the ACMA assay, the rates measured for KirBac3.1 fall into a physiologically relevant range.

**Disulfide-locked pores are too narrow for spermine entry**. To gauge the degree of width restriction afforded by disulfide linkages, and hence the relative plasticity at the Tyr132 collar, we utilized an open-channel blocker specific to Kir channels (spermine) and compared its binding in native and disulfide-linked pores using two methods, MD and functional assay. Spermine reversibly blocks activated Kir channels in a voltage-dependent manner, entering from the cytosol and occluding the conduction pathway[15,16]. The response to membrane depolarization is conceptually similar to the outward motion of conserved arginine side chains in the voltage sensors of $K_V$ channels[43]. A crystal structure of KirBac3.1 with spermine bound shows the extended polyamine in the cavity with its tail end protruding into the cytoplasm through a snug opening at the Tyr132 collar[18]. In the presence of spermine, the opening is ~1 Å wider than in wild type, the incremental expansion being induced by spermine entry.

To determine whether spermine was able to bind and block conduction under the assay conditions, the polyamine was applied to proteoliposomes containing disulfide-linked A133C-T136C. In contrast to the inhibitory effect of spermine on wild-type KirBac3.1, spermine failed to inhibit $K^+$ conduction through the disulfide-linked pore, indicating that it struggled to penetrate the Tyr132 collar to reach its blocking site within the conduction pathway (Fig. 5a). This suggests the narrowest (transverse) dimension of spermine is wider than the disulfide-linked cytoplasmic opening. It also indicates that the permeating $K^+$ species is only partially hydrated, as fully hydrated $K^+$ is bulkier than spermine (Fig. 5b).

Steered MD and umbrella sampling simulations were employed to estimate the free-energy required to move intracellular spermine into the pore cavity. Figure 5c plots the position of a central methylene carbon, C8, of spermine, as it moves from the cytosol into the cavity, against the PMF. The

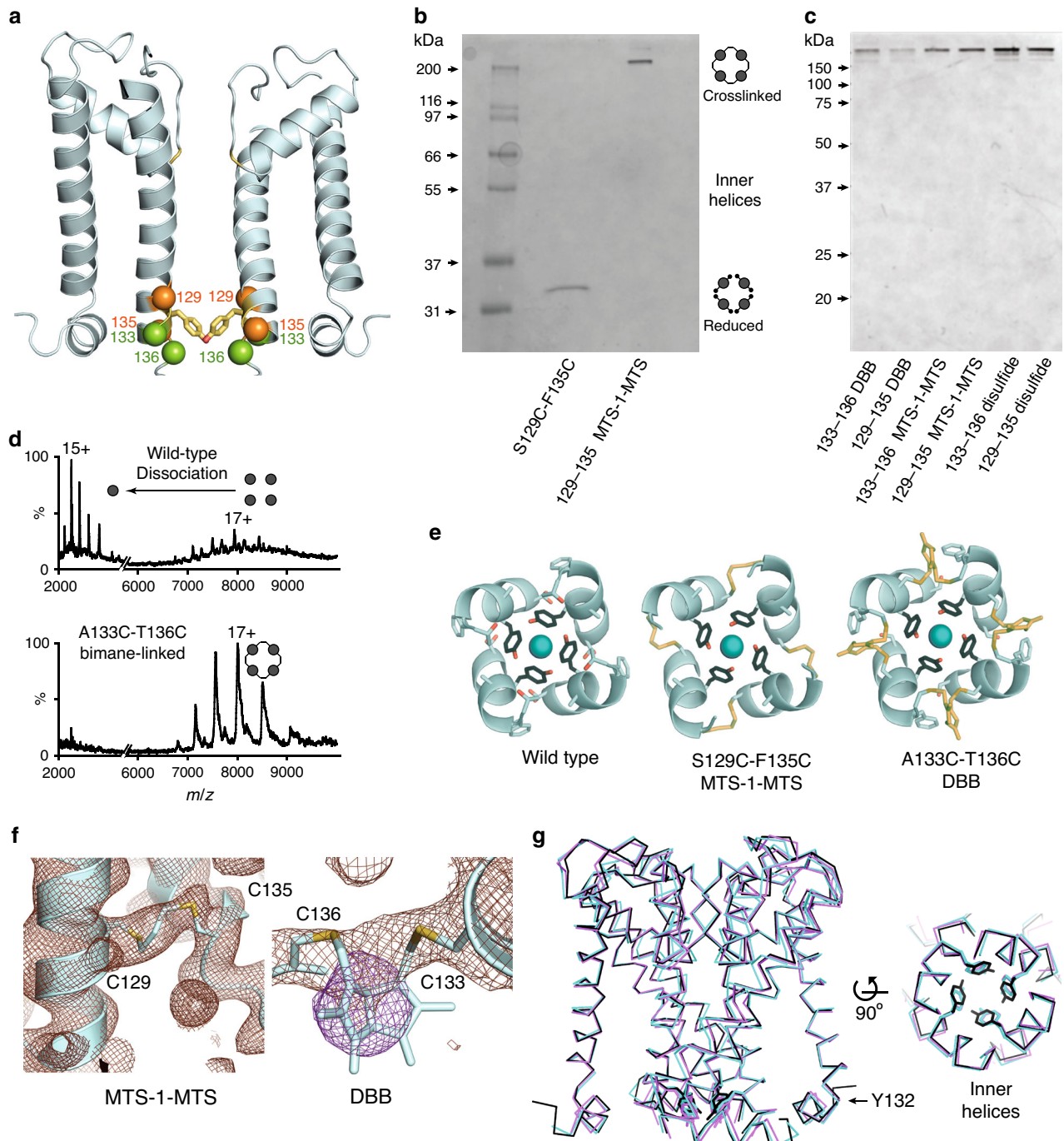

**Fig. 3 Crosslinking around Tyr132 forms covalent tetramers and limits inner helix movement. a** Sites of the paired cysteine mutations relative to Tyr132, with Cα shown as spheres. Sites of 129 and 135 are colored orange, and 133 and 136 shown in green. For clarity, only two subunits are depicted. **b** In the presence of crosslinker, the cysteine-pair mutant S129C-F135C MTS-1-MTS migrates on SDS-PAGE as a tetramer, whereas reduced S129C-F135C migrates as a monomer. Both samples are of reconstituted proteoliposomes and are representative of at least $n = 20$ gels. **c** All six mutant-crosslinker combinations migrate as tetramers and are representative of at least $n = 10$ gels. **d** Native mass spectrometry using collision-induced dissociation in the gas phase results in loss of signal for wild type and reduced cysteine-pair mutants, but not for covalently crosslinked mutants. Spectra for wild-type (upper) and dibromobimane-linked A133C-T136C (lower). **e** Comparative cross-sectional slices through crystal structures at the Tyr132 collar. Only the inner helices are shown. Crosslinkers are depicted as yellow sticks and $K^+$ ions as deep cyan spheres. **f** Electron density for the crosslinkers is present at all subunit interfaces of S129C-F135C (MTS-1-MTS) and A133C-T136C (dibromobimane). Left panel shows a 3.1 Å composite omit map ($2|F_o| - |F_c|$) contoured at 1.5 σ and showing strong density for the bridging S-CH$_2$-S moiety. Right panel shows a simple 4.1 Å $2|F_o| - |F_c|$ omit map contoured at 1 σ (brown mesh), overlaid with an unsharpened $|F_o| - |F_c|$ omit map contoured at 3 σ (purple mesh) revealing the bimane moiety. **g** Superposition of Cα traces of crosslinked structures S129C-F135C MTS-1-MTS (cyan) and A133C-T136C DBB (purple) onto the native pore structure (black). Two subunits have been omitted for clarity in the left-hand view.

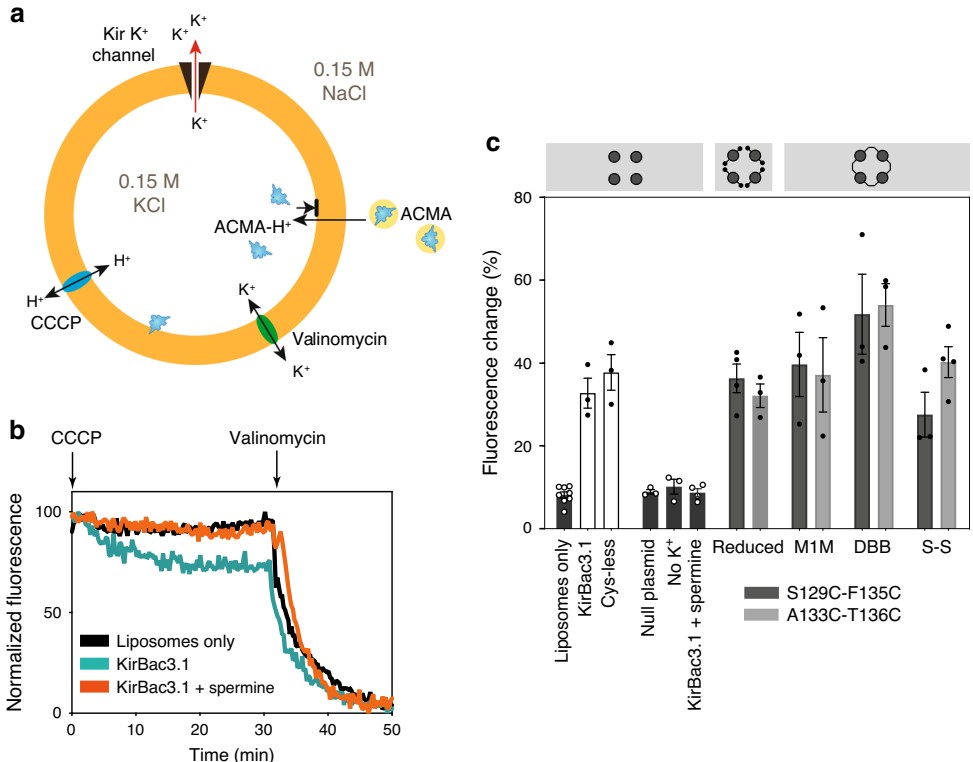

**Fig. 4 Potassium permeates through crosslinked KirBac3.1. a** A lipid-soluble proton-sensitive dye (9-amino-6-chloro-2-methoxyacridine, ACMA) is equilibrated with proteoliposomes (orange circle) prior to addition of a protonophore (carbonyl cyanide m-chlorophenylhydrazone, CCCP). Protons influxing through CCCP bind ACMA, which manifests as a decrease in fluorescence emission and is balanced by $K^+$ efflux through the $K^+$ channel. Protonated ACMA cannot pass out through the membrane. Limiting fluorescence is determined by addition of the $K^+$ ionophore valinomycin. The total fluorescence change measured in the assay is summed from individual proteoliposomes. **b** Normalized raw data from wild-type channels. **c** Summary of potassium flux experiments. Data shown as mean ± SEM ($n = 8$ (channel-free)), 3 (wild-type, Cys-less, null plasmid, no $K^+$, A133C-T136C reduced, S129C-F135C MTS-1-MTS, A133C-T136C MTS-1-MTS, S129C-F135C dibromobimane, A133C-T136C dibromobimane, S129C-F135C disulfide), 4 (wild-type spermine block, S129C-F135C reduced, A133C-T136C disulfide experiments); details of replicates are in Supplementary Table 4. Each black or white circle represents the mean from an individual experiment. Liposomes only = reconstituted with buffer-only; KirBac3.1 = wild type; Cys-less = KirBac3.1 (C262S, C71V, C119V); No $K^+$ = ($Na^+_i$/$NMDG^+_o$/ionophore= monensin); KirBac3.1 + spermine ($500_i$/$1000_o$ μM). M1M = mts-1-mts; DBB = dibromobimane; S–S = disulfide.

resultant PMF values indicate a significantly greater energetic barrier to spermine at the Tyr132 collar in disulfide-linked A133C-T136C than in wild type (unconstrained), with respective maxima at site 2 of 15 kJ $mol^{-1}$ for wild type and 25 kJ $mol^{-1}$ for the disulfide-linked mutant. The 10 kJ $mol^{-1}$ difference between them infers ~50-fold lower probability of the spermine passing the disulfide-linked constriction of the cysteine-pair mutant relative to the unconstrained channel, whereas 15 kJ $mol^{-1}$ difference at site 3 approximates a 400-fold lower probability of it reaching that site. The approximate differences in PMF at positions 1 to 4 are 5, 10, 15 and 45 kJ $mol^{-1}$, respectively, with the disulfide-linked mutant always the higher of the two.

In wild-type channels at zero field, the energetic barrier of 15 kJ $mol^{-1}$ faced by spermine as its leading amine N1 passes the Tyr132 collar (Fig. 5c; 1–>2) is the same as that experienced as N2 passes the tyrosine collar (3–>4). At sites 1 and 3, and prior to spermine engaging the channel, the energy is approximately equal. In contrast, in the disulfide-locked pore, there is a significant barrier at site 4 above the PMF at site 3, 40–45 kJ $mol^{-1}$, preventing spermine from penetrating much beyond the barrier at Tyr132. The MD-predicted differences in free-energy of spermine binding at serial sites between wild-type and disulfide-linked A133C-T136C KirBac3.1 accord with the experimental assay results that spermine blocks wild type but not disulfide-linked channels.

Probability density profiles corresponding to the position of spermine in the transmembrane cavity at zero field, and 25 and 50 mV $nm^{-1}$ (Fig. 5d) illustrate the impact of the 15 kJ $mol^{-1}$ barrier to spermine penetration in wild-type channels at different field strengths. At zero field, there is only a small probability that spermine traverses the tyrosine collar and reaches site 3, whereas at 25 and 50 mV $nm^{-1}$ progressively more spermine penetrates past the constriction at Tyr132. The calculations indicate that the probability of spermine penetrating further into the pore increases with the field strength applied, in accord with the known voltage-dependence of polyamine block[16]. Sites 1 and 3 exhibit near equivalence in energy under these simulation conditions (noting that ions are absent from the upper cavity).

## Discussion
Although the conventional view that gating entails switching from a narrow to a wide conformation has proved challenging to verify, studies on BK[44,45] and K2P[46,47] channels have indicated that conformational change of $K^+$ channel pores during gating is not universal. Underpinning the canonical gating model is the idea that the intracellular opening to the pore must be sufficiently wide to accommodate a flow of hydrated $K^+$ ions during conduction. Data presented here suggest that the natural limiting width for permeation is closer to the ionic diameter of $K^+$ and that the requirement for substantial conformational change

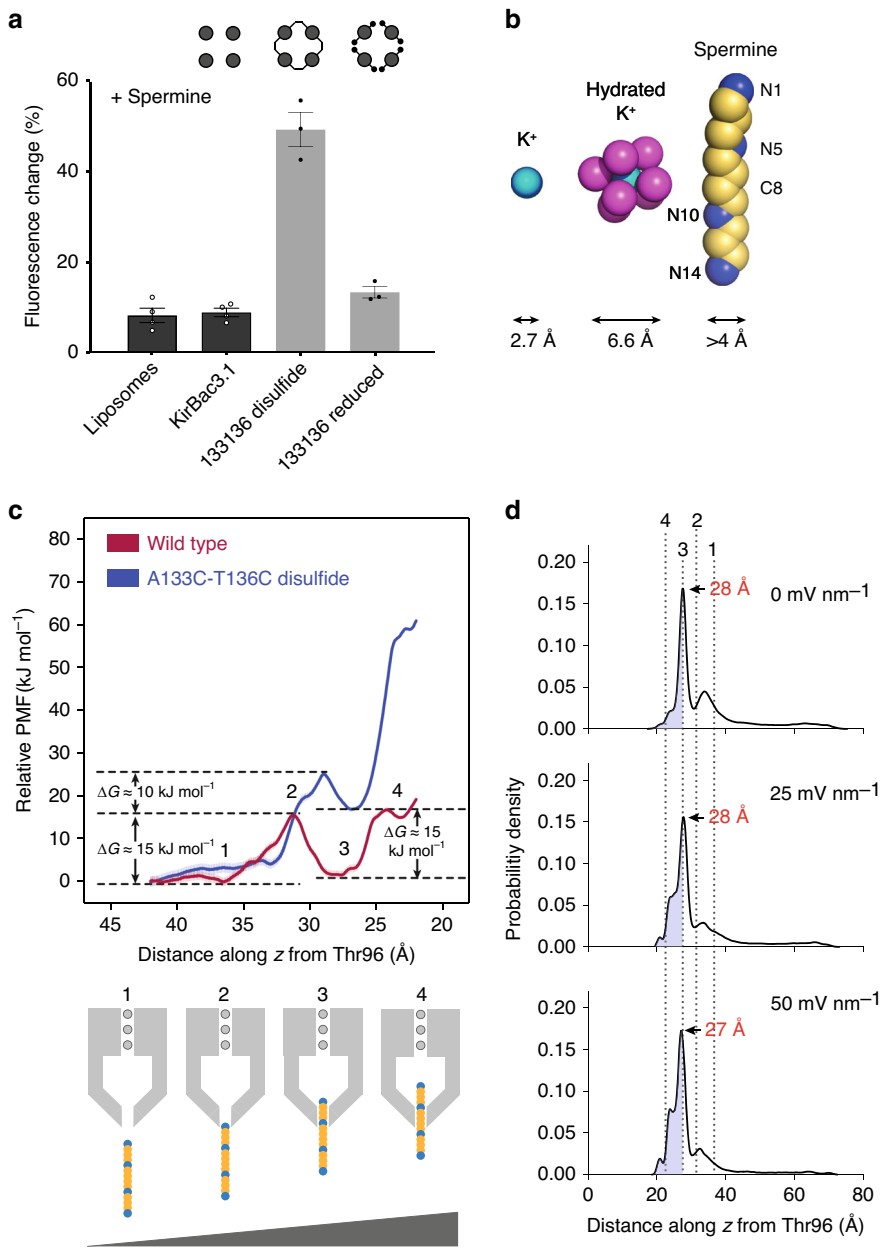

**Fig. 5 Spermine enters wild type but not disulfide-linked pores. a** Summary of potassium flux experiments investigating spermine block in wild-type and disulfide-linked A133-T136C channels. Data are shown as mean ± SEM (*n* = 4 (channel-free, wild type)) or 3 (A133C-T136C disulfide, A133C-T136C reduced experiments); details of replicates are in Supplementary Table 4. Each black or white circle represents the mean of an individual experiment. **b** Comparative dimensions of permeant cations (K⁺ is depicted in deep cyan, oxygen in magenta, carbon in yellow and nitrogen in blue). **c** The PMF represents the energetic barrier to spermine block. Positions 1 to 4 refer to the positioning of a middle carbon (C8) of spermine relative to the pore—shown in the schematic below. **d** The voltage-dependency of spermine block is illustrated by the trend in probability density of spermine entering the pore at field strengths of 0, 25 and 50 mV nm⁻¹. The shaded region of each panel corresponds to 80% of structures in the MD simulations, shown relative to positions 1–4. The distance of C8 relative to the center of mass of Thr96 is annotated for major peaks (red). In the region 18.0–27.5 Å, the occupancy sequentially changes from ~25 to 40–50% with increasing field (0–25 to 50 mV nm⁻¹). The field-free distribution of spermine in the cavity predicts only a small probability of spermine occupancy at position 4.

within the pore is obviated by the propensity of K⁺ for ligand exchange, analogous to carbonyl exchange in the selectivity filter. The MD simulations on KirBac3.1 indicate that the coordination shell of K⁺ is fleetingly depleted as it passes through an intracellular opening that, whereas smaller than the diameter of a hydrated K⁺ ion, is sufficient to accommodate partially hydrated ions. We verified this in functional assays where the width of the opening was covalently constrained, finding that this did not

prevent K⁺ passing through and demonstrating that the constriction is ineffective at stemming K⁺ flux. To verify the covalent bonds linking subunits together were effective at preventing pore dilation at the collar, we carried out spermine block experiments on the disulfide-linked (most tightly constricted) mutant, A133C-T136C. The inability of spermine to block the crosslinked, but not the reduced, mutant verifies the covalent disulfide linkages are effective at preventing relative movement of the inner helices.

Potassium conduction through K⁺-selective channels requires multiple transfers between coordination shells. As K⁺ enters the selectivity filter its hydration shell is exchanged for peptide carbonyl oxygen atoms; these are serially exchanged as it transfers between the four binding sites and to then be replaced by water molecules as K⁺ exits the selectivity filter. Although ion dehydration represents a significant contribution to the free-energy barriers governing ion permeation, it cannot be considered in isolation; dipole moments, environmental permittivity (dielectric) and the energetics of hydration within a protein cavity of limited size will differ from the energetics in the selectivity filter or the hydrogen-bonded network of bulk water. The large ionic radius of K⁺ relative to other biological cations means its hydration shell is less strongly bound and its relative dehydration energy lower. The energetic costs incurred by transient partial dehydration may be offset by favorable interactions with the protein, including dipoles in the polar environment. Work on gramicidin A indicates that K⁺ can travel several Angstroms through very narrow pores[29,48,49]. The gramicidin A pore accommodates water in single file across the lipid bilayer, and one-dimensional PMF simulations indicate K⁺ can pass through with only two coordinating waters, the positive charge stabilized by the alignment of dipoles from water and polar groups of the protein[29]. In contrast, the distance through the hourglass-like constriction at Tyr132 of KirBac3.1 is very short, as in human Kir2 channels, affording ions almost simultaneous access to water molecules on either side of the opening (Fig. 6 and Supplementary Movie 1) to replenish their hydration shell as they pass through. A recent study demonstrating that K⁺ favors cation–π over cation–water interactions[50], suggests that in KirBac3.1 (and in all human Kir3 and Kir6 subfamily members) a weakly bound hydration shell and the aromatic rings at the constriction are particularly favorable for partial dehydration.

Our data demonstrate that, in Kir channels, control over ion flux is not explained by the canonical pore-gating model of conformational change and that, although some K⁺ channels conduct through very wide pores, the width is not essential; the limiting aperture for conduction is effectively set by the ionic diameter of K⁺ (<3 Å). Interestingly, a recent study noted spontaneous transition of K⁺ from bulk to cavity through a narrow aperture, which is also inconsistent with the constriction acting as a gate[51]. There are several possible alternatives to the canonical model. In BK[45] and K2P[47] channels, the idea that gating occurs at the selectivity filter is gaining in acceptance as a way to explain how a wide pore might deactivate without narrowing. In Kir channels a selectivity filter gate may also have relevance[18], but in the opposing context of a narrow constriction being unable to prevent K⁺ flow.

The molecular ramifications of lipid binding provide a logical point from which to consider mechanism. In eukaryotic Kir channels, the anionic lipid PI(4,5)P₂ regulates activity and is essential to both activation and conduction[22,52,53], but PI(4,5)P₂ does not appear to bring about pore widening; how it enables conduction is thus of particular interest. Conversely, long chain CoAs antagonize Kir2 activity[54]. From another perspective, changes at the subunit interfaces of the tetrameric intracellular assembly in Kir channels have been implicated in gating[18,28,52]. Findings presented here highlight a further possible determinant: an absolute requirement for free water molecules on either side of the shallow bottleneck (Fig. 6). As it passes through the narrow opening, the hydration shell of K⁺ is transiently depleted. Any inability to replenish it on the other side of the opening would inflate the free-energy cost of passage, increasing resistance to conduction. A source of water molecules at the intracellular entrance is therefore essential but, even within the aqueous medium of the cell, this cannot be taken for granted in Kir

channels as their intracellular assembly cloisters the pore entrance. Interestingly, defined conformational changes at the subunit interfaces effect molecular changes at or near the membrane surface that may have broader relevance for gating and conduction. The changes include natural siphons for water movement to or from the pore entrance that could also control water access to emerging ions.

## Methods

**Experimental objectives and study design**. The objectives of the study were to investigate (and cross validate) the conformational requirements of gating and conduction in inward rectifier potassium channels. Molecular dynamics were employed to evaluate the conformational changes and free-energy barriers to ions passing along the conduction pathway of KirBac3.1. Structure analysis and native mass spectrometry were used to validate recombinant purified mutant channels in which the dimensions of the pore were constrained by covalent linkages. The function of crosslinked mutants reconstituted into liposomes was evaluated by an established fluorimetric assay.

**Materials**. Detergents were purchased from Anatrace, lipids from Avanti Polar Lipids and all other reagents, unless otherwise stated, from Sigma Aldrich. Kirbac3.1 mutant cDNA was purchased from DNA2.0. Chromatography columns and resins are from GE Healthcare.

**Protein preparation**. Kirbac3.1 was expressed in C41(DE3) (Lucigen. Cat# 60442) or LEMO21(DE3) (New England Biolabs, Cat# C2528J) E. coli at 18 °C for 18 h, or at 16 °C for 72 h, respectively. Cells were harvested and resuspended in 20 mM TRIS pH 8.0 and 150 mM KCl containing EDTA-free Complete Protease Inhibitor (Roche) followed by lysis by two passes through a Stansted homogenizer (Stansted Fluid Power) at 100 MPa. Membranes were harvested by centrifugation at 265,000×g for 40 min at 4 °C and the protein solubilised in 100 ml of 20 mM TRIS pH 8.0, 150 mM KCl, 1 mM PMSF and 1% (w/v) 3,12-Anzergent for 45 min. Insoluble material was removed by centrifugation. Imidazole to 20 mM was added to the supernatant, which was loaded on a 5 ml IMAC HiTrap column (GE) charged with Co²⁺. Protein was eluted with a linear gradient of imidazole, up to 500 mM, in buffer containing 20 mM TRIS pH 8.0, 500 mM KCl, 0.1 mM PMSF and 0.05% (w/v) dodecylmatoside (DDM). Fractions containing KirBac3.1 were concentrated in a 100 kDa molecular weight cutoff centrifugal concentrator (Millipore) and applied to a Superdex 200 10/30 column (GE) equilibrated with 20 mM TRIS pH 8.0, 150 mM KCl, 0.02% (w/v) DDM and 0.5 mM TCEP. Fractions corresponding to the KirBac3.1 tetramer were concentrated with a 100 kDa centrifugal concentrator (Millipore) to between 10 and 40 mg ml⁻¹, snap-frozen in liquid N₂ and stored at −80 °C as single use aliquots.

**Crosslinking**. KirBac3.1 mutants were diluted to 0.03–4 mg ml⁻¹ (depending on the crosslinker) in crosslinking buffer (20 mM TRIS pH 8.0, 150 mM KCl and 0.02% (w/v) DDM) and crosslinked by addition of MTS-1-MTS, dibromobimane or CuSO₄ to 0.5 mM, 2.5 mM and 1 mM, respectively, followed by incubation at room temperature for 30 min. EDTA was added to 5 mM to the CuSO₄ cross-linking reaction and MMTS to 100 mM to the MTS-1-MTS and dibromobimane reactions. Excess crosslinking reagents were removed with Micro Bio-Spin 6 columns (BioRad) equilibrated in crosslinking buffer. The crosslinked tetramer was isolated from higher molecular weight species by size exclusion chromatography with a Superdex 200 10/30 column (GE) equilibrated with crosslinking buffer. Fractions corresponding the crosslinked KirBac3.1 tetramer were recovered and concentrated with a 100 kDa centrifugal concentrator (Millipore) to between 10 and 20 mg ml⁻¹, snap-frozen in liquid N₂ and stored at −80 °C as single use aliquots, until required.

**Crystallization and data collection**. KirBac3.1 and crosslinked KirBac3.1 mutants at a concentration of 2.3 to 4 mg ml⁻¹ were crystallized in sitting drops by vapor diffusion at the Bio21 C3 crystallization facility in Parkville against 2.5% PEG 4k; 2.5% PEG 8k; 10–17% PEG 400; 90 mM HEPES pH 7.5 1 mM TCEP and 50 mM EDTA or 10 mM CaCl₂ for reduced Kirbac3.1, 33% (v/v) PEG 400; 0.1 M MES; pH 6.5; 4% (v/v) ethylene glycol; 0.1 M NaCl for dibromobimane crosslinked KirBac3.1 A133C-T136C and 13% (w/v) PEG MME 2000; 0.1 M TRIS-HCl; pH 7.1; 0.1 M CaCl₂ and 1% (w/v) LDAO for MTS-1-MTS crosslinked KirBac3.1 S129C-F135C. All crystals were cryoprotected prior to data collection by the addition of TMAO pH 8.0 to 2 M. All crystallographic data were collected at beamline MX2 at the Australian Synchrotron using Blu-Ice[55].

**Structure determination and refinement**. Crystallographic data were processed and scaled using HKL2000[56] or processed and scaled using XDS[57] and reduced using the CCP4 suite[58]. Electron density in all structures was phased by molecular replacement, using *PHASER* as implemented in the CCP4i suite[59]. A molecular replacement model for the wild-type structure was derived from previous protein data bank entries from our group and the 2.0 Å refined model (wild type) reported

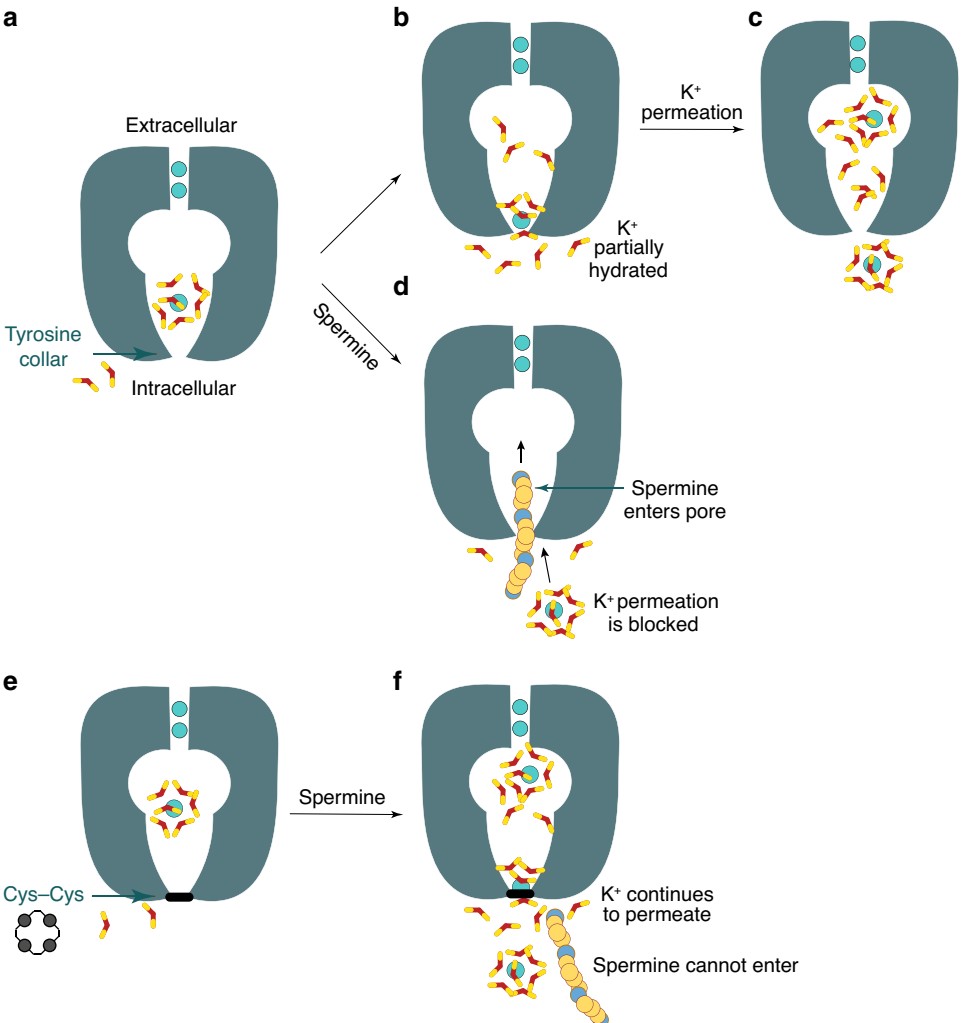

**Fig. 6 Schematic representation of the findings.** **a** A hydrated K$^+$ (cyan sphere) within the wild-type Kir pore has a hydration shell of 6–7 water molecules (red and yellow sticks). **b** As the pore narrows to a bottleneck, the ion partially dehydrates by shedding 3–4 water molecules. **c** The hydration shell of K$^+$ is replenished from the other side of the opening. **d** A spermine molecule penetrates into the cavity of the wild-type channel, preventing outward ion conduction. **e** When the Tyr132 collar is disulfide-linked, **f** K$^+$ permeates but spermine cannot enter the conduction pathway.

here used as a model for the remaining structures. Maps for the two crosslinked mutant structures were B-factor sharpened[60]. Model building was carried out using COOT[61] and refinement using PHENIX[62] except for the 4.0 Å refinement of the bimane-crosslinked A133C-T136C mutant, which used target structure restraints[63] in BUSTER-TNT[64]. Atomic coordinates were iteratively refined by maximum likelihood and simulated annealing procedures applied, alternating with cycles of TLS and individual B-factor refinement. Occupancy was refined for ligands and residues with alternate conformations. Refinement was monitored according to the decrease in $R_{free}$. A number of residues in each structure were omitted or had side chains truncated at Cβ due to positional disorder. Statistics are presented in Supplementary Table 1. The structures have been deposited in the Protein Data Bank with accession codes 6O9T, 6O9U and 6O9V.

**Structural comparison.** Superposition of the S129C-F135C-MTS-1-MTS structure onto a 2.0 Å native KirBac3.1 structure shows a close fit of the polypeptide backbone of the entire assembly; the RMS deviation calculated for coordinate displacement of ~1136 Cα positions (residues 12–295) is 1.1 Å, whereas for the pore alone (residues 33–136) it is 0.6 Å. The fit of the A133C-T136C-bimane structure, in which a rigid pyrazole-based linkage connects residue 133 on one subunit to residue 136 on an adjacent, to the 2.0 Å native shows minimal distortion of either pore or cytoplasmic assembly (which were superimposed separately because of a relative interdomain 'twist' rotation[18]). The RMS deviation calculated for coordinate displacement over the transmembrane pore (residues 33–136) is 0.6 Å and for the cytoplasmic assembly (residues 137–295) is 1.1 Å.

**Molecular dynamics.** The 2.0 Å crystal structure of native KirBac3.1 (PDB 6O9U) was used as an initial model for simulations. The disulfide-bonded model was

derived from this template by introducing cysteine substitutions at residues 129 and 135 or 133 and 136 of each subunit via homology modeling using MODELLER v9.14 software[65]. The whole simulation system, including the formation of disulfide bonds between adjacent inner helices, was prepared using the CHARMM-GUI Membrane Builder server[66], with N- and C-terminal residues patched as standard termini. Transmembrane pores (residues 33–138) of native and disulfide-linked channels were separately embedded within a standard POPC bilayer, each containing 162 lipids, using the replacement method. In addition, each starting model contained four potassium ions: three sites within the selectivity filter (S1, S3, S4) and one in the internal cavity (near the selectivity filter). One additional potassium ion or spermine molecule was placed in the lower cavity for potassium or spermine calculations respectively. Spermine was placed into channel cavity at the same position observed in structure 2WLK. Twenty-nine water molecules were also placed in the cavity for potassium calculations and 35 for spermine calculations. The simulation cells comprised ~60,000 atoms, of which approximately 30,000 were water molecules, in a box of dimensions $8.2 \times 8.2 \times 8.8$ nm$^3$. TIP3P water parameters were used in order to solvate all systems and all ionizable residues were assumed to be in their dominant protonation states at pH 7. Sufficient K$^+$ and Cl$^-$ ions were introduced by replacement of water molecules to bring the systems to an electrically neutral state at an ionic strength of 0.15 M. The principal axis of the pore was aligned along the z axis.

All simulations were performed using the GPU-accelerated GROMACS software package (version 2018.3)[67] and CHARMM36m force field[68]. CMAP corrections were applied for all components; all atoms including hydrogen atoms were included. To obtain the equilibrated channel-ion-solvent-membrane complex, an elongated CHARMM-GUI Membrane Builder preparation protocol was followed. Steepest descent energy minimization was followed by six sequential steps of equilibration with a gradual decrease in the restraining force applied to different

components. After 101.5 ns NVT and NPT equilibrations, the resulting structures were utilized in steered molecular dynamics (SMD). The LINCS algorithm was applied for resetting constraints on covalent bonds to hydrogen atoms, which allowed 2 fs time steps for MD integration during the entire simulation. The particle-mesh Ewald algorithm was used for calculating electrostatic interactions within a cutoff of 12 Å, with the Verlet grid cutoff-scheme applied for neighbor searching, using an update frequency of 20 and a cutoff distance of 12 Å for short-range neighbors. A 12 Å cutoff was applied to account for van der Waals interactions, using a smooth switching function starting at 1.0 nm. Periodic boundary conditions were utilized in all directions. During the equilibration stages, the temperature was maintained at 303.15 K using a Berendsen-thermostat with a time constant of 1.0 ps. Protein, membrane and ion-water groups were treated independently to increase accuracy. The pressure was maintained at 1.0 bar by semi-isotropic application of a Berendsen-barostat, with a time constant of 5.0 ps. During steered molecular dynamics, the temperature was maintained at 303.15 K using a Nose–Hoover-thermostat with a time constant of 1.0 ps, with protein, membrane and ion-water groups treated independently, with the pressure maintained at 1.0 bar using the Parrinello–Rahman-barostat semi-isotropically, with a time constant of 5.0 ps and compressibility of $4.5 \times 10^{-5}$ bar$^{-1}$. The umbrella sampling method was used to calculate the potential of mean force (PMF) when ions passed through the Tyr132 collar. The reaction coordinate of a potassium ion was defined relative to the center-of-mass (COM) of the four Thr96 residues at the base of the selectivity filter along the $z$ direction. The initial structures for umbrella sampling were extracted from the 120 ns long umbrella pulling SMD and the subsequent 100 ps long NPT equilibration. Pulling potentials were set for two discrete pulling groups: the potassium ion in the lower cavity and four Thr96. The pulling geometry was direction-periodic, with a harmonic force constant during umbrella pulling of 3000 kJ mol$^{-1}$ nm$^{-2}$. The rate of change was 0.15 Å ns$^{-1}$ in each case. Although subjecting a target ion to pulling forces, a flat-bottomed position restraint was applied to the ion to restrain it within a cylinder with a radius of 2 Å running parallel to the $z$ axis. The initial structures for umbrella sampling were extracted from the 120 ns long umbrella pulling SMD, and subsequently equilibrated for 100 ps. Each umbrella sampling window was simulated for 200 ns, with a total of 97 or 115 individual umbrella sampling windows involved in each instance, equivalent to ~20 μs of molecular dynamics simulations. The windows were centered at ~0.2 Å intervals along the $z$ axis, covering the range from 16.5 to 36.5 Å relative to the COM of the four Thr96 residues; during umbrella sampling the flat-bottomed position restraint was removed. A static external electric field with a strength of 0.05 V nm$^{-1}$ was applied along the membrane normal direction toward the cytoplasmic domain side in steered molecular dynamics and umbrella sampling. Final PMF profiles were calculated using gmx-wham and the statistical uncertainties estimated via Bayesian bootstrap analysis with 200 bootstraps and a tolerance of $10^{-6}$.

The number of oxygen atoms within a prescribed cutoff distance of 3.0 Å of the target ion was calculated using VMD[69]. Two-dimensional histograms were constructed using the ggplot2 package in the R software package. An unbiased distribution was derived from biased distributions along reaction coordinate and the PMFs from WHAM analysis. 2D-PMFs were created using WOLFRAM MATHEMATICA 11.2.0.0.

Further unrestrained simulations were carried out on the wild-type pore in an electric field at 50 or 100 mV nm$^{-1}$; in each, 20 simulations were run over 50 ns (total of 1 μs at each holding potential). The interchangeable diagonals were statistically analyzed as D1 and D2 couplets for 100,001 structures extracted at 10 ps intervals from each 1 μs trajectory. All hexbin figures were symmetrized by exchanging the indices of diagonals; that is, the couplets of (D1, D2) were duplicated in the form (D2, D1). All probability density functions were calculated using the ggplot2 package in R software and integration calculations were performed using the Grace plotting tool.

For the spermine calculations, the same simulation procedures were observed, except that the reaction coordinate was defined as the distance of C8 of spermine relative to the center-of-mass of the four Thr96 residues. Spermine was steered in the physiological direction from the bulk solution toward the lower cavity. To generate a sensible sampling ensemble during the steered process, distance restraints between opposite and adjacent oxygen atoms of Tyr132-hydroxyl group were applied and the distances between oxygen atoms were restrained in a range from 8 to 20 Å and 5.7 to 14.1 Å for opposite and adjacent pairs, respectively. The distance restraints were removed for the ensuing umbrella sampling simulations. There were 106 and 126 umbrella sampling windows for wild-type and disulfide-linked A133C-T136C, respectively.

Further unrestrained simulations were carried out on the wild-type pore in an electric field at 0, 25 or 50 mV nm$^{-1}$; in each, 20 simulations were run over 50 ns (total of 1 μs at each holding potential). The interchangeable diagonals were statistically analyzed as D1 and D2 couplets for 100,001 structures extracted at 10 ps intervals from each 1 μs trajectory. All hexbin figures were symmetrized by exchanging the indices of diagonals; that is, the couplets of (D1, D2) were duplicated in the form (D2, D1). The distribution of C8 atoms of spermine were statistically analyzed, for those frames in which the spermine passed the periodic boundary, those values of distance were converted to positive values by adding the instantaneous length in $z$ direction of box. All probability density functions were calculated using the ggplot2 package in R software and integration calculations were performed the Grace plotting tool.

**KirBac liposome reconstitution**. Lipids (50% 1-palmitoyl-2-oleoyl-sn glycero-3-phosphocholine:POPC, 5% 1-palmitoyl-2-oleoyl-sn-glycero-3-phospho-L-serine: POPS, 45% total E. coli lipids:TECL in the ratio POPC:TECL:POPS 50:45:5) were dissolved in chloroform:methanol (65:35), dried to a thin film under N$_2$ and residual solvent removed by lyophilization overnight. Water was added to a final lipid concentration of 25 mg ml$^{-1}$ followed by incubation at 37 °C for 1 h and 5 freeze/thaw cycles. An equal volume of 2× inside buffer (IB) was added to give a final buffer composition of 20 mM Tris-HCl pH 8.5 and 150 mM KCl followed by extrusion to 400 nm (two rounds of 11 passages through the filter) in a mini-extruder (Avanti Polar Lipids) preheated to 65 °C. In some experiments, NaCl was substituted for KCl.

For reconstitution, dodecyl-β-maltopyranoside (DDM) was added to a 1:8 molar detergent:lipid ratio to liposomes and incubated for 15 min at room temperature followed by the addition of protein to a protein:lipid molar ratio of 1:555 and incubation for 1 h at room temperature.

Excess detergent was removed by passing 80 μl of proteoliposomes through G50 resin (settled bed volume 3 ml) equilibrated in inner buffer (IB), followed by elution with IB. Further detergent removal was accomplished by dilution of liposomes to a final volume of 1.5 ml by centrifugation at $91,000 \times g$ for 1 h at 25 °C. The liposome pellet was resuspended in 100 μl IB, centrifuged at $3000 \times g$ for 3 min and any precipitate discarded. Liposome size uniformity was assessed using dynamic light scattering (Zetasizer—Malvern Instruments) and protein reconstitution by SDS-PAGE using known concentrations of KirBac3.1 in DDM as standards. Amount of protein added was optimized using wild-type KirBac3.1 to achieve an average of 0.3 functional channels per liposome, with equivalent protein: lipid ratios used for the crosslinked experiments.

**Liposomal fluorimetric assay**. Fluorimetric liposome assays were performed within 24 h of KirBac reconstitution. Each 5 μl aliquot of prepared liposomes was diluted to a final volume of 110 μl in outside buffer (OB: 20 mM Tris-HCl pH 8.5 and 150 mM NaCl), resulting in a final buffer composition of 20 mM Tris-HCl pH 8.5, 143.2 mM NaCl and 6.8 mM KCl. In some experiments, the outer buffer (OB) contained 150 mM NMDG$^+$ instead of NaCl. The pH sensitive dye 9-amino-chloro-2-methoxyacridine (ACMA) was added to a final concentration of 2 μM from a freshly prepared 200 μM stock in 100% (v/v) ethanol. Fluorescence emission at 483 nm (excitation at 419 nm) was monitored over time with path lengths of 2 and 10 mm for excitation and emission, respectively. Baseline fluorescence was monitored for 3 min before the proton specific ionophore carbonyl cyanide m-chlorophenyl hydrazine (CCCP) was added ($t = 0$) to a final concentration of 9 nM from a freshly prepared 500 nM stock prepared in OB. A decrease in fluorescence is observed when the membrane is permeable to K$^+$. After 35 minutes ($t = 35$), the K$^+$ specific ionophore was added to 20 nM from a freshly prepared 2 μM stock made up in 100% (v/v) ethanol. The assay was allowed to reach a final electro-chemical equilibrium over a further 20 minutes. In some experiments, the K$^+$ channel blocker spermine was added to a final concentration of 500/1000 μM in IB/OB before the establishment of the fluorescence baseline. Control liposomes were prepared identically to proteoliposomes with the exception of the addition of an equal volume of crosslinking buffer in place of protein.

All fluorescence experiments were normalized to the fluorescence measured at $t = 0$, which represents maximal fluorescence (100%); and the fluorescence measured after the addition of valinomycin, which represents the minimal fluorescence where all liposomes have equilibrated (0%). A first-order exponential decay was fitted from $t = 0$ to $t = 35$ to determine the change in fluorescence due to K$^+$ flux through reconstituted KirBac channels.

**Mass spectrometry**. Protein samples were buffer exchanged into 200 mM ammonium acetate pH 8.0, 0.5% C8E4 using Biospin-6 (BioRad) columns prior to mass spectrometry analyses and were directly introduced into the mass spectro-meter using gold-coated capillary needles prepared in-house[70]. Data were collected on both a modified quadrupole-time-of-flight (Q-ToF) mass spectrometer[71] (Waters, Manchester, UK) and a modified 5 QExactive hybrid quadrupole-Orbitrap mass spectrometer[72] (Thermo Fisher Scientific Inc., Berman, Germany) optimized for analysis of high mass complexes. The instrument parameters used for data collection on Q-ToF were: capillary voltage 1.55 kV, cone voltage 200 V, extractor 5 V, collision voltage 400 V, backing pressure 5.6 mbar and argon col-lision gas pressure 0.2 MPa, and for collection on QExactive were: capillary voltage 1.2 kV, S-lens RF potential 100 V, quadrupole selection from 2000 to 20,000 $m/z$ range, collisional activation in the HCD cell 200 V, argon pressure in the HCD cell $1.12 \times 10^{-9}$ mbar, resolution of the instrument 17,500 at $m/z = 200$ (a transient time of 64 ms). Calibration of the instruments was performed using 10 mg ml$^{-1}$ solution of cesium iodide in water. CID experiments were performed on the above-mentioned Q-ToF instrument. Since, the Kir channel is a membrane protein introduced in a detergent micelle, 200 V of activation energy was used to colli-sionally activate the protein-detergent complex. This was necessary to remove detergent micelles. An additional 200 V energy was then applied to perform the CID experiments. Species present in all the crosslinked samples were verified on high resolution QExactive instrument and measured masses are listed in Supple-mentary Table 2.

**Liposomal flux assay**. Each liposomal flux experiment was performed across at least three independent reconstitution experiments. Each independent measurement was the result of averaging between two and six technical replicates. To assess significance, a two-sided Dunnett's multiple comparisons test was carried out to compare each experimental condition to either channel-free liposomes or wild-type KirBac3.1 proteoliposomes. Significance was determined by a family-wide threshold of 0.05, with the $P$-value for each comparison adjusted for multiplicity. For each experimental condition, the number of independent (and constitutive technical) replicates and $P$-values as determined by the Dunnett's multiple comparisons are summarized in Supplementary Table 3.

**Reporting summary**. Further information on research design is available in the Nature Research Reporting Summary linked to this article.

## Data availability

Data supporting the findings of this manuscript are available from the corresponding authors upon reasonable request. A reporting summary for this Article is available as a Supplementary Information file. Atomic coordinates and structure factors have been deposited in the Protein Data Bank with accession codes PDB 6O9T, PDB 6O9U, PDB 6O9V. Source data underlying Figs. 4c and 5a are provided as a Source Data file.

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

## Acknowledgements

We thank Mike Lawrence for advising on low-resolution crystallography. We also thank Peter Colman, Jamie Vandenberg, Matthew Call, Sam el-Ajouz, Pauline Crewther, Agalya Periasamy and Xiaowen Xiao for help and advice. Diffraction data were collected on the MX2 beamline at the Australian Synchrotron, Victoria, Australia. Crystallization trials were carried out at the C3 Parkville Facility by Janet Newman and Shane Seabrook. The project was in part funded by NHMRC Australia Project Grants 1006624 and 1080682, an NHMRC Senior Research Fellowship (J.M.G.), and discretionary funding from The Walter and Eliza Hall Institute. K.A.B. was supported by an Australian Postgraduate Award. R.J. and S.H. acknowledge receipt of Australian Research Training Scholarships. The work in C.V.R group is supported by Medical Research Council Grant MR/N020413/1. Part of this work was undertaken using resources from the National Computational Infrastructure, which is supported by the Australian Government and provided through Intersect Australia Ltd, and through the HPC-GPGPU Facility, which was established with the assistance of LIEF Grant LE170100200. The work was made possible through Victorian State Government Operational Infrastructure Support and Australian Government NHMRC IRIISS.

## Author contributions

K.A.B. and D.M.M. planned and carried out protein biochemistry and assays and crystallized the protein. B.J.S., R.J. and S.H. designed and performed all molecular simulations. J.R.B. and C.V.R. devised and performed mass spectrometry experiments. D.L., P.J., M.W. and A.P.H. contributed to experimental design and interpretation. J.M.G., O.B.C., D.M.M. and K.A.B. carried out crystallography. C.J.B. was responsible for reagent synthesis. J.M.G. conceived the project, guided the experimental work and wrote the manuscript.

## Competing interests

The authors declare no competing interests.
