## [Peer Review File · Nature Communications]

Peer Review File - Reviewers' comments first round:

Reviewer #1 (Remarks to the Author):

The manuscript by Black et al investigates the conformational states required for gating and conduction in the K⁺ channel protein Kir. The authors use a combination of techniques including molecular dynamics simulations and functional assays to provide insights into how K⁺ permeates the channel and interestingly how H₂O-coordination of K⁺ changes during the process. Through the use of cysteine paired point mutants, the authors selectively crosslinks four pore domain subunits of Kir to constrain and attenuate dimensions of the intracellular mouth of the channel and provide evidence that a widening of the channel is not required for function.

As requested and as a non-expert on K⁺ channels, I have primarily performed a technical assessment of the experimental evidence the authors use to characterize their important cross-linked Kir mutants. In my view, the authors convincingly show by the use of SDS-PAGE (reduced vs native) that the generated Kir mutants forms tetramers after crosslinking (Figure 3b and c). The authors further support this finding by performing native MS and gas-phase induced unfolding by CID to show that the native tetrameric Kir dissociates into individual subunits whereas the crosslinked tetrameric Kir does not. While the spectra in Figure 3d do appear to support this conclusion, the presentation lacks experimental details e.g. 1) was a wildtype Kir tetramer observed by native MS in the absence of CID and 2) what collision energy was used to record the two spectra in 3d, presumably this was identical?

Finally, what evidence do the authors have that the Kir subunits are cross-linked in the right places i.e. the right Cys residues? The authors are strongly encouraged to perform a peptide mapping analysis of the 6 cross-linked variants to confirm that crosslinking has occurred in the right places.

Overall, I think the manuscript is interesting, well-written and nicely integrates results from several techniques to support the main conclusions. From the above described technical viewpoint, I recommend publication of the manuscript with minor changes as described.

Reviewer #2 (Remarks to the Author):

The authors describe a combined experimental and computational study with as main finding that a significantly opened helix bundle crossing is not a requirement for potassium permeation in Kir potassium channels. Rather, the relatively small opening observed at the cytoplasmic side in crystal structures of Kir channels is shown to be sufficient to allow the efficient permeation of partially dehydrated potassium ions. The manuscript is well written and interesting. However, I have a number of concerns about both the novelty of the findings as well as the computational part of the work.

First, a recent MD study (DOI:10.1085/jgp.201912359) of Kir3.2 presented a PMF showing a higher barrier for potassium permeation at the selectivity filter than at the activation gate, implying that the restriction at the activation gate is not rate limiting. Although the opening at the activation gate is not the main focus of that paper, it still takes some of the novelty of the current work. This should be discussed.

Second, although the focus of the current work is on the activation gate, the link of the simulation work to the rest of the paper would be a lot stronger if overall permeation through the channel (including the selectivity filter, SF) would be reported. Currently, only partial permeation (through the activation gate) has been demonstrated. Extrapolation to the overall permeation process implies assuming that the activation and inactivation (SF) gates are independent. However, there is ample evidence of strong coupling between the two. Therefore, without simulation of complete permeation events the partial permeation events across the activation gate only provide limited information.

Third, I am concerned about the spermine PMFs (Fig. 5d). First, reading the value from the very steep curves as done is associated with substantial uncertainties, as thermal fluctuations render

the exact binding location uncertain. In addition, the ~ 20 kJ/mol higher spermine free energy for WT at the crystallographic site, as compared to bulk, outside, is worrying. If it costs 20 kJ to go from bulk to the crystallographic position, then spermine would not spontaneously bind at the crystallographic position?

Finally, the discussion reads "the pore must be sufficiently wide to accommodate a flow of hydrated K^+ ions during conduction, yet data presented here suggest K^+ ions can readily pass through much smaller openings than this." However, this is not a novel finding of the current study. Since the first structure of KcsA it is known that K^+ ions can readily pass the narrow selectivity filter. Therefore, it would seem more appropriate to note that the (partial) dehydration occurring at the activation gate is perhaps not unlike the permeation at the selectivity filter in this respect.

Reviewer #3 (Remarks to the Author):

Black et al. try to answer the question of whether a key region along a Kir's pore (as observed in crystal structures) is too narrow, or it is actually sufficiently wide, to conduct K^+ ions, with multiple approaches: Crystallography, MD simulation, cross-linking, and flux assays. They propose that, if K^+ ions would undergo a transient and partial dehydration process, then these partially hydrated ions could pass through the narrow part of the pore that fully hydrated K^+ ions could not.

Their proposal would help explain how K^+ ions could move through a pore, part of which is as constricted as it is in thus far available crystal structures. Unfortunately, the studies underlying the proposal are insufficient. To answer the question, one first needs to know the experimental rate of K^+ ions moving through the pore, and then demonstrates that the proposed mechanism can quantitatively account for the rate, neither of which has yet been provided by the authors.

We thank all three reviewers for their constructive comments and careful review of the manuscript.

Reviewer #1 (Remarks to the Author):

The manuscript by Black et al investigates the conformational states required for gating and conduction in the K⁺ channel protein Kir. The authors use a combination of techniques including molecular dynamics simulations and functional assays to provide insights into how K⁺ permeates the channel and interestingly how H₂O-coordination of K⁺ changes during the process. Through the use of cysteine paired point mutants, the authors selectively crosslinks four pore domain subunits of Kir to constrain and attenuate dimensions of the intracellular mouth of the channel and provide evidence that a widening of the channel is not required for function.

As requested and as a non-expert on K⁺ channels, I have primarily performed a technical assessment of the experimental evidence the authors use to characterize their important cross-linked Kir mutants. In my view, the authors convincingly show by the use of SDS-PAGE (reduced vs native) that the generated Kir mutants forms tetramers after crosslinking (Figure 3b and c). The authors further support this finding by performing native MS and gas-phase induced unfolding by CID to show that the native tetrameric Kir dissociates into individual subunits whereas the crosslinked tetrameric Kir does not. While the spectra in Figure 3d do appear to support this conclusion, the presentation lacks experimental details e.g. 1) was a wildtype Kir tetramer observed by native MS in the absence of CID and 2) what collision energy was used to record the two spectra in 3d, presumably this was identical?

We thank the reviewer for their favourable comments on our crosslinking experiments. Since, the Kir channel is a membrane protein introduced in a detergent micelle, 200 V of activation energy was used to collisionally-activate the protein-detergent complex. This is necessary to remove detergent micelles. An additional 200 V energy was then applied to perform the CID experiments. Spectra shown in figure 3d were recorded under identical conditions (i.e. with a total energy of 400 V). The additional details are now provided in our revised version (Methods).

Finally, what evidence do the authors have that the Kir subunits are cross-linked in the right places i.e. the right Cys residues? The authors are strongly encouraged to perform a peptide mapping analysis of the 6 cross-linked variants to confirm that crosslinking has occurred in the right places.

We had initially attempted to introduce Cys mutations on a native background and identified the issues that have been raised by the reviewer. In the studies reported in this manuscript, all three native Cys residues have been removed by substitution ('cysteine-less' background), and cysteine pairs introduced at the specific points noted. Each subunit thus has only two cysteine residues; in one mutant these are sited at residues 133 and 136, and in the other at residues 129 and 135. The only Cys crosslinks residues possible are between these paired residues. This was validated by mass spectrometry – the mass differences provided in Extended Data Table 3 verify the presence of four crosslinkers per tetramer.

Overall, I think the manuscript is interesting, well-written and nicely integrates results from several techniques to support the main conclusions. From the above described technical view-point, I recommend publication of the manuscript with minor changes as described.

Reviewer #2 (Remarks to the Author):

The authors describe a combined experimental and computational study with as main finding that a significantly opened helix bundle crossing is not a requirement for potassium permeation in Kir potassium channels. Rather, the relatively small opening observed at the cytoplasmic side in crystal structures of Kir channels is shown to be sufficient to allow the efficient permeation of partially dehydrated potassium ions. The manuscript is well written and interesting. However, I have a number of concerns about both the novelty of the findings as well as the computational part of the work.

We would like to address the novelty of the findings up front. The focal point of the manuscript is conformational change at the constriction, which is the basis of the mechanism of canonical potassium channel gating as described in biology textbooks and taught at a tertiary level. *Our finding that the constriction is ineffective at gating K⁺ currents, and hence that conformational widening is not essential to permit current, is unprecedented.*

The assumption that a constricted pore prevents conduction of K⁺ has not previously been challenged, *i.e.* ALL present models assume K⁺ cannot permeate when the helix bundle crossing is narrower than the diameter of hydrated K⁺ ions, irrespective of any other gating elements within the conduction pathway.

We believe that our direct analysis of the putative ‘activation gate’ will have a tremendous impact on research into ion channels. The canonical model underpins experimental design in research, as well as the interpretation of structural and functional outcomes and yet, as we have now demonstrated, does not stand up to scrutiny. Also, most (all?) published MD studies on narrow-pore channels (*e.g.* Kir/KcsA) adopt the view that there IS a conformational change at the helix bundle constriction (including DOI:10.1085/jgp.201912359); often an artificially widened structural model is used as the starting point for simulations, and sometimes constrained throughout (*e.g.* Heer et al, eLife, 2017, <https://doi.org/10.7554/eLife.25844.001>). Both decisions can potentially influence the MD outcomes.

First, a recent MD study (DOI:10.1085/jgp.201912359) of Kir3.2 presented a PMF showing a higher barrier for potassium permeation at the selectivity filter than at the activation gate, implying that the restriction at the activation gate is not rate limiting. Although the opening at the activation gate is not the main focus of that paper, it still takes some of the novelty of the current work. This should be discussed.

We appreciate the reviewer drawing our attention to an interesting recent MD study. To preface our comments, we note that in the MD study canonical steric gating at the helix bundle crossing was assumed, and thus the starting point in the majority of these simulations was derived from a single MD structure (selected because the constriction had widened), rather than seeded from randomised structures. However, we were impressed that in a small proportion of the MD runs they fortuitously observed spontaneous transition of K⁺ from bulk to cavity through an aperture that had narrowed, consistent with the outcomes of our assays which reveal that the constriction at the helix bundle crossing is ineffective as a gate. *We have referenced the work in the main body of text as: “Interestingly, a recent study noted spontaneous transition of K⁺ from bulk to cavity through a narrow aperture, consistent with our finding that a constriction is ineffective as a gate”.*

Their finding that the barrier in the selectivity filter, ~20 kJ mol⁻¹, through which all ions must readily pass, is significantly higher than the barrier at the constriction, supports our argument that the constriction does not hinder conduction. A direct comparison of PMF energies between the studies is not possible due to subtle differences in methodology. In DOI:10.1085/jgp.201912359, PMF energy barriers were calculated solely from K⁺ resident frequencies without allowing for other contributing factors. Moreover, while our PMF values are independent of field strength, theirs are dependent on field strength and combine observations from simulations performed at different field strengths to generate PMFs. We have noted in the text at line 216: “Interestingly, a recent study noted spontaneous transition of K⁺ from bulk to cavity through a narrow aperture, which is inconsistent with the constriction acting as a gate (ref: Bernsteiner, JGP, 2019).”

Second, although the focus of the current work is on the activation gate, the link of the simulation work to the rest of the paper would be a lot stronger if overall permeation through the channel (including the selectivity filter, SF) would be reported. Currently, only partial permeation (through the activation gate) has been demonstrated.

While this is a reasonable point, other published MD studies have looked at overall permeation and it was not the focus of this study. Ever since the comparison of KcsA and MthK by McKinnon (Jiang et al 2002 Nature 417, 523), the helix bundle crossing has received a great deal of attention as the ‘activation gate’ and our study is unique in experimentally testing it directly. As such, it focuses on the constriction in question, not permeation across the entire channel. Our findings clearly show, for the first time, that the gating elements responsible for activation of Kir channels must be located elsewhere than the helix bundle

crossing; this is a paradigm shift. To resolve the issue of gating mechanism will require an entirely different combination of simulation and experiment and is our present focus.

We have therefore altered the title and abstract to clarify the scope of the study.

Title: “A constriction at the inner helix bundle of Kir channels does not impede conduction of potassium ions”.

Abstract: “The reversible transition distinguishing conducting and non-conducting states of K⁺ channels is conventionally explained by a conformational change causing the intracellular entrance to the conduction pathway to dilate sufficiently to accommodate fully hydrated K⁺ ions. While studies have established that some classes of K⁺ channel can gate even while a wide intracellular mouth is maintained, the view that K⁺ cannot permeate a constriction at the helix bundle crossing narrower than the diameter of hydrated K⁺ ions has remained unchallenged. To explore this, we tested the function of narrow inward rectifier K⁺ (Kir) channels that had the intracellular mouth locked in a conformation too narrow to accommodate fully hydrated ions. Function was unimpaired in these channels. In parallel, we used an all-atom molecular dynamics approach to simulate K⁺ ions moving along the conduction pathway between the lower cavity and the intracellular entrance. During simulations, the constriction at the inner helix bundle crossing did not significantly widen. Instead, transient partial dehydration facilitated K⁺ permeation past the constriction at the helix bundle crossing. The low free energy barrier to partial dehydration and a lack of largescale conformational change indicate Kir channels are not gated by the canonical mechanism.”

Extrapolation to the overall permeation process implies assuming that the activation and inactivation (SF) gates are independent. However, there is ample evidence of strong coupling between the two. Therefore, without simulation of complete permeation events the partial permeation events across the activation gate only provide limited information.

We concur with the reviewer that interdependent actions in distinct regions of the pore are likely to synergise in pore-gating but reiterate that the focus of the paper is on the canonical K⁺ channel ‘gate’, where the cysteine-pair mutations are located. In large part, the basis of coupling between inner and outer regions of the pore has remained elusive because present interpretation of structure-function relationships is strongly influenced by the canonical model of gating at the helix bundle crossing.

Third, I am concerned about the spermine PMFs (Fig. 5d). First, reading the value from the very steep curves as done is associated with substantial uncertainties, as thermal fluctuations render the exact binding location uncertain.

We appreciate the constructive comment, and in its wake have revisited the spermine PMFs and carried out steered MD and umbrella sampling simulations on wild type and 133-136 disulfide-crosslinked channels, in which the opening is very tightly constrained. Spermine block is a voltage-dependent process, one that has been attributed, in part, to the movement (or displacement of) permeant by impermeant ions in the membrane field (Spasova and Lu, 1998, JGP, 112, 211-221). With this in mind, we removed a potassium ion that was occupying the upper cavity region in our original calculations and reperformed all of the spermine simulations (~50 μsec simulation time in all). Removing this ion permitted spermine binding in the lower region of the cavity, indicating that potassium ions in the upper region of the conduction cavity influence spermine binding by altering the free energy of binding. This reveals a new aspect of spermine block and one we had not previously appreciated. In these new calculations, we modelled spermine egress in the physiological direction, moving from the cytosol into the cavity. The energetic barrier in wild type channels under zero-field conditions is only 15 kJ mol⁻¹ as the first amine passes the side chain of Tyr132; the total energy returns to near zero immediately the first two nitrogen amines pass the Tyr132 sidechain and are solvated in the cavity, consistent with spermine binding and blocking. In the disulfide-crosslinked channel the same barrier is slightly higher (at ~25 kJ mol⁻¹) than the wild type. However, with spermine bound at stage ③ in the cavity, the energy is 15 kJ mol⁻¹ higher, consistent with a significantly greater barrier to spermine penetration than in wild type.

We have combined panels **c and d** of Figure 5 and replaced with the new data (now Fig. 5c).

The text now reads: "Steered MD and umbrella sampling simulations were employed to estimate the free energy required to move intracellular spermine into the pore cavity. Figure 5c plots the position of a central methylene carbon, C8, of spermine against the PMF, as it moves from the cytosol into the cavity. In line with the assay results, the resultant PMF values indicate a significantly greater energetic barrier to spermine penetration into the cavity in disulfide-linked A133C-T136C than in wild type. The energetic barrier faced by spermine as N1 passes the Tyr132 collar and, simultaneously, N2 approaches the tyrosine hydroxyls (*i.e.* Fig. 5c ①->②) of 15 kJ mol⁻¹ is the same as that experienced by N2 as it passes the tyrosine collar and N3 engages the hydroxyls (*i.e.* Fig. 5c ③->④), where ④ is approximately the crystallographic position."

In addition, the ~20 kJ/mol higher spermine free energy for WT at the crystallographic site, as compared to bulk, outside, is worrying. If it costs 20 kJ to go from bulk to the crystallographic position, then spermine would not spontaneously bind at the crystallographic position?

Our original simulations were carried out in the absence of an applied field in order to determine the free energy barrier to permeation faced by spermine. However, spermine binding is voltage-dependent and so to address the reviewer's question we have calculated probability density profiles corresponding to the pore at zero field, and 50, and 100 mV nm⁻¹. The outcomes indicate that the probability of the spermine penetrating further into the pore increases with the field strength applied. This is shown in Fig. 5d.

The text now reads " Probability density profiles corresponding to spermine entering the transmembrane cavity at zero field, and 50 and 100 mV nm⁻¹ (Fig. 5d) illustrate the impact of the 15 kJ mol⁻¹ barrier to spermine penetration in wild type at different field strengths. At zero field, there is only a small probability that spermine traverses the tyrosine collar, while at 100 mV nm⁻¹ the highest probability corresponds to spermine adopting the crystallographic position. These calculations indicate that the probability of spermine penetrating further into the pore increases with the field strength applied, in accord with the known voltage-dependence of polyamine block."

The methods have also been altered in accordance.

Finally, the discussion reads "the pore must be sufficiently wide to accommodate a flow of hydrated K⁺ ions during conduction, yet data presented here suggest K⁺ ions can readily pass through much smaller openings than this." However, this is not a novel finding of the current study. Since the first structure of KcsA it is known that K⁺ ions can readily pass the narrow selectivity filter. Therefore, it would seem more appropriate to note that the (partial) dehydration occurring at the activation gate is perhaps not unlike the permeation at the selectivity filter in this respect.

We thank the reviewer for their insight. Early structures revealed the narrow aperture of the selectivity filter and highlighted the need for the ions to shed their coordination shell of water molecules in order to enter it. We concur with the reviewer that the two processes are analogous, both being allowed by the propensity of K⁺ for ligand exchange. Where they diverge is that within the selectivity filter, the carbonyl oxygens act as surrogates for displaced water molecules solvating the K⁺ ion. In contrast, at the helix bundle constriction there is no obvious provision to compensate for displaced water. Thus, while there are parallels between our findings and serial ligand exchange at the selectivity filter, it has been widely assumed that the energy cost of dehydration in the absence of surrogate carbonyl ligands is prohibitive. Our data show unequivocally that this is not the case.

We agree that the sentences in question does not convey our message adequately and have revised accordingly. It now reads: "...the pore must be sufficiently wide to accommodate a flow of hydrated K⁺ ions during conduction. Data presented here suggest that the natural limiting width for permeation is closer to the ionic diameter of K⁺ and that the requirement for substantial conformational change within the pore is obviated by the propensity of K⁺ for ligand exchange, analogous to carbonyl exchange in the selectivity filter."

Reviewer #3 (Remarks to the Author):

Black et al. try to answer the question of whether a key region along a Kir's pore (as observed in crystal

structures) is too narrow, or it is actually sufficiently wide, to conduct K⁺ ions, with multiple approaches: Crystallography, MD simulation, cross-linking, and flux assays. They propose that, if K⁺ ions would undergo a transient and partial dehydration process, then these partially hydrated ions could pass through the narrow part of the pore that fully hydrated K⁺ ions could not.

Their proposal would help explain how K⁺ ions could move through a pore, part of which is as constricted as it is in thus far available crystal structures. Unfortunately, the studies underlying the proposal are insufficient. To answer the question, one first needs to know the experimental rate of K⁺ ions moving through the pore, and then demonstrates that the proposed mechanism can quantitatively account for the rate, neither of which has yet been provided by the authors.

We thank the reviewer for their thoughtful remarks. ACMA-based assays are widely recognised as sensitive but ‘slow’, and not amenable to determination of physiological rates. In using an ACMA assay to show functional reconstitution of the voltage-dependent proton channel, H_v1, Rod Mackinnon qualified “*The fluorescence-based assay, which indirectly measures H⁺ flux through an unknown relationship between H⁺ concentration and fluorescence, precludes quantitative determination of H⁺ conduction rates*” (Lee et al, JMB, 2009, 387, 1055-1060), later echoed by Benjamin Gerdes (Anal. Bioanal. Chem. 2018. 410, 6497-6505). While rates *can* be determined by single channel recordings, the method was deemed inappropriate for measuring function in our case because of the need to verify that crosslinking is complete (each of the mutant-crosslinker combinations required individual optimisation before reconstitution). In other words, in single channel recordings in cells or bilayers it would not be possible to state with certainty that measurable currents were not due to a tiny amount (ppm) of non-crosslinked channel in the sample. A population assay, as used here, circumvents this potential pitfall, because a tiny fraction of individual incompletely crosslinked channels do not significantly impact the outcomes.

We can, however, address the assay rates in a ‘relative’ manner by comparison of our KirBac3.1 data with studies on its close homologue Kir3.2 (GIRK2): *Glaaser and Schlessinger (2017) Sci. Rep. 7, 4592* and *Lacin et al (2017) J. Gen. Physiol., 8, 799*, which provide useful points of reference. Capitalising on the fact that Kir3.2 requires bound PIP(4,5)P₂ to enable conduction, these studies were able to present comparative fluorimetric traces from ACMA assays on Kir3.2 measured in the presence of different types or concentrations of regulatory phosphoinositide lipids. Importantly, ‘fractional’ activation by titration was demonstrated by variation in the nominal conduction rate according to PIP reagent or concentration. In Glaaser and Schlessinger, the rate varied with the concentration of PI(4,5)P₂(8:0); the relative rate of K⁺ efflux (1/tau) calculated in a comparable ACMA assay being ~0.003 s⁻¹ at the midpoint of the sigmoidal curve of rate x [PIP₂(8:0)], (with an EC₅₀ of ~25 μM). This demonstrated that although ACMA assays do not measure absolute rates, the internal consistency reflects our current knowledge of the biology. The rates calculated from our experimental curves of KirBac3.1 fall in the mid-range of those reported for Kir3.2, indicating the KirBac3.1 conduction rate is comparable to Kir3.2 and thus physiologically relevant.

To demonstrate this point, we have taken the liberty of attaching Fig. 2 panel b from Glaaser and Schlessinger below, plotting a red filled circle onto the curve. The red circle marks the the average rate of conduction observed across the wild type and KirBac3.1 mutants – both crosslinked and reduced.

The rates (s^{-1}) and experimental time constants (τ) from KirBac3.1 have been included as Extended Data Table 5. The differences between crosslinked and reduced forms of the pore are not significant, which indicates the enforced constriction is not rate limiting. We note that this is in accord with the point made by Reviewer 2 that if the calculated barrier at the constriction is lower than calculated at the selectivity filter it cannot be rate limiting.

We have inserted some lines at the end of the section “Crosslinked Kir mutants mediate K^+ flux in fluorimetric liposomal assays”.

“The differences in rate and overall signal between crosslinked and reduced forms of the pore are not significant, which indicates the enforced constriction is not rate limiting. The rates (s^{-1}) and experimental time constants (τ) have been included as Extended Data Table 5.

While ACMA assays do not replicate the fast physiological rates of ion channel function, published titration studies demonstrate their internal consistency⁴¹. The rates calculated from our experimental curves of KirBac3.1 fall in the mid-range of rates reported for Kir3.2 titrations under very similar conditions⁴¹, indicating that in the context of the ACMA assay, the rates measured for KirBac3.1 fall into a physiologically relevant range.”

Reviewers' comments second round:

Reviewer #1 (Remarks to the Author):

The authors have adequately addressed my comments.

Reviewer #2 (Remarks to the Author):

Most of my concerns have been satisfactorily addressed. However, my concern about the spermine binding remains. The new Figure 5d unfortunately does not help to resolve this. Rather, it shows a correspondence between simulation and the crystallographic position only for simulations with an applied voltage of 100 mV/nm, whereas the crystals were presumably grown in the absence of such a voltage? Therefore, it seems misleading to write, as the revised manuscript now reads: "These calculations indicate that the probability of spermine penetrating further into the pore increases with the field strength applied, in accord with the known voltage-dependence of polyamine block." as it highlights the correspondence with voltage-dependence of polyamine block but does not mention the discrepancy between crystallography and simulation.

Related, the magnitude of the applied voltage in the simulations is questionable. Assuming that the reported 100 mV/nm is applied across the simulation box, and assuming that the simulation box size is on the order of 10 nm in the applied field direction, an applied voltage of on the order of 1V would be active. This is a lot higher than any (electro)physiological range. Therefore, this questions the validity of the asserted "accord with the known voltage-dependence of polyamine block."

Reviewer #3 (Remarks to the Author):

This reviewer appreciates the authors' responses to the earlier comments. The central point of the manuscript is captured by the title: "A constriction at the inner helix bundle of Kir channels does not impede conduction of potassium ions." This extraordinary claim is still not backed up by necessary data.

At 150 mM K⁺ used in the study, a K⁺ channel typically conducts K⁺ at a rate $10^6 - 10^8 \text{ M}^{-1} \text{ s}^{-1}$. In order to substantiate the authors' claim, high quality single-channel electric-current recordings (or an alternative, adequate method) from individual channels, which are known to have the constriction, are absolutely necessary.

The flux assay used is not an adequate substitute for the electric-current recordings. The K⁺ flux rate of the examined channels estimated from the assay is several orders of magnitude lower than the expected K⁺ permeation rate (Fig. 4b). Regarding the cited data of Glaaser and Schlessinger, the highest flux rate is also extremely low, $< 0.01 \text{ s}^{-1}$. It appears that a process, other than K⁺ conduction, in the assay limits the observed rate. Thus, at best, any data that have thus far been obtained with this method could not address the issue of whether a constriction at the inner helix bundle of Kir channels impedes conduction of K⁺. Should the apparent kinetics of the flux assay reflect that of K⁺ permeation, the observation of the extremely slow kinetics would strongly reject the notion that the ion conduction was not impeded.

Reviewer #1 (Remarks to the Author):

The authors have adequately addressed my comments.

We are pleased the reviewer is satisfied and thank them for their insight and application.

Reviewer #2 (Remarks to the Author):

Most of my concerns have been satisfactorily addressed. However, my concern about the spermine binding remains. The new Figure 5d unfortunately does not help to resolve this. Rather, it shows a correspondence between simulation and the crystallographic position only for simulations with an applied voltage of 100 mV/nm, whereas the crystals were presumably grown in the absence of such a voltage? Therefore, it seems misleading to write, as the revised manuscript now reads: "These calculations indicate that the probability of spermine penetrating further into the pore increases with the field strength applied, in accord with the known voltage-dependence of polyamine block." as it highlights the correspondence with voltage-dependence of polyamine block but does not mention the discrepancy between crystallography and simulation.

We appreciate the constructive comments made by Reviewer #2. The reviewer implicitly agrees that the statement "These calculations ..." is correct in that it highlights the correspondence with voltage-dependence and polyamine block, however, we accept that superficially it does not resolve an apparent discrepancy between the X-ray crystal structure and the simulations (at zero applied field). The calculations at zero field indicate that a small population, but not the majority, of spermine does penetrate deep into the cavity (density beyond ~ 23 Å, Figure 5d, top panel), corresponding with the crystal structure. The excess spermine over protein in the reported crystallisation conditions (approximately 1,000-fold molar excess) would push the equilibrium distribution to the left of this panel, bringing experiment and theory into concordance.

There is thus no significant discrepancy between our simulation results and the original X-ray crystal structure.

We have revised the caption for figure 5 to alert the reader to the origins of the apparent discrepancy: "(d) The voltage-dependency of spermine entry is illustrated by the trend in probability density of spermine entering the pore at field strengths of 0, 25 and 50 mV nm⁻¹. The shaded region of each panel corresponds to 80% of structures in the MD simulations, shown relative to the position of the same atom in crystal structures. The distance of C8 relative to the centre of mass of Thr96 is annotated for major peaks (red). In the region 18.0 to 27.5 Å, the occupancy sequentially changes from approximately 25 to 40 to 50% with increasing field (0 to 25 to 50 mV nm⁻¹). The field-free distribution of spermine in the cavity predicts only a small probability of spermine occupancy at the experimentally observed position (C8 ~ 23 Å). A significant (~ 1000 -fold) molar excess of spermine over protein in the crystallisation conditions¹⁸ is more than sufficient to account for the ~ 15 kJ mol⁻¹ energy difference needed to populate position ④, corresponding to the observed X-ray crystallographic position of spermine."

Related, the magnitude of the applied voltage in the simulations is questionable. Assuming that the reported 100 mV/nm is applied across the simulation box, and assuming that the simulation box size is on the order of 10 nm in the applied field direction, an applied voltage of on the order of 1V would be active. This is a lot higher than any (electro)physiological range. Therefore, this questions the validity of the asserted "accord with the known voltage-dependence of polyamine block."

This reviewer raises a very important issue affecting calculations that apply a constant electric field across the simulation system. Across the literature there are mixed approaches, applying different field strengths, measured across different regions of the simulation (the selectivity filter, the transmembrane region or across the simulation system); for example:

- Jensen *et al.*, Principles of conduction and hydrophobic gating in K⁺ channels. *Proc. Natl. Acad. Sci. USA*, (2010) 107, 5833-5838. Voltages of -180 mV < V < 180 mV were applied across the selectivity filter; the length of the SF was measured as 1.34 nm, which implies a field strength of ~130 mV/nm.
- Kopec *et al.*, Molecular mechanism of a potassium channel gating through activation gate-selectivity filter coupling. *Nat. Commun.* (2019) 10, 5366. Simulations of MthK channel where an electric field was applied to generate a membrane voltage of ~300 mV.
- Gumart *et al.*, Constant electric field simulations of the membrane potential illustrated with simple systems. *Biochim. Biophys. Acta – Biomembranes* (2012) 1818, 294-302. Here a voltage of 500 mV to 1 V was applied across the system; Roux used potentials of this magnitude to illustrate the effectiveness of the constant electric field approach.
- Treptow *et al.*, Coupled Motions between Pore and Voltage-Sensor Domains: A Model for *Shaker B*, a Voltage-Gated Potassium Channel. *Biophys. J.* (2004) 87, 2365-2379. Applied a field 5-times larger than experimental conditions, a 500 mV transmembrane voltage, to promote a faster response from the system.
- Ulmschneider *et al.*, Molecular dynamics of ion transport through the open conformation of a bacterial voltage-gated sodium channel. *Proc. Natl. Acad. Sci. USA*. (2013) 110, 6364-6369. Applied fields of between 4 and 75 mV/nm, corresponding to transmembrane potentials of 39 to 665 mV.

These highlight a troublesome grey area, *that there is very little consistency or agreement in the application of electric field in the reports presently in the literature*. In view of this, we sought evidence of a trend, rather than absolute values.

The profiles in figure 5d illustrate the correspondence between voltage-dependence and polyamine block, illustrating that more spermine is able to penetrate the cavity more deeply as the membrane potential increases. While the potentials, 0, 50 and 100 mV nm⁻¹ were selected to clearly show the trend, we acknowledge that the data at 100 mV/nm are unlikely to be physiological. We have therefore removed the 100 mV nm⁻¹ data and replaced it with data from simulations at 25 mV nm⁻¹. This new data reaffirms the trend at lower, more physiologically relevant, potentials.

To more accurately reflect the situation, we have replaced the manuscript text (page 6) “At zero field, there is only a small probability that spermine traverses the tyrosine collar, while at 100 mV nm⁻¹ the highest probability corresponds to spermine adopting the crystallographic position.” with “At zero field, there is only a small probability that spermine traverses the tyrosine collar and reaches the crystallographic position, while at 25 and 50 mV nm⁻¹ progressively more spermine penetrates deeply into the cavity.”

At the close of the paragraph, we have also added the sentence “The near equivalence in energy of sites ① and ③ means a spermine molecule that penetrates as far as ③ is as stable as one penetrating as far as site ①.”

Reviewer #3 (Remarks to the Author):

This reviewer appreciates the authors' responses to the earlier comments.

The central point of the manuscript is captured by the title: "A constriction at the inner helix bundle of Kir channels does not impede conduction of potassium ions." This extraordinary claim is still not backed up by necessary data.

While our finding is unprecedented, it is based on an unbiased treatment of controlled experimental data, and in no sense an ‘extraordinary claim’.

The number of water molecules coordinating K⁺ ions is not fixed, nor does K⁺ only coordinate water. Coordination preference is influenced by microenvironment and chemistry (*e.g.* there is a thermodynamic preference of K⁺ for cation-pi interactions over water coordination, cited in the manuscript). Thus, there is no objective reason to assume that K⁺ ions cannot pass through a significantly narrower aperture at the helix bundle crossing (Tyr132) than has been hitherto assumed.

Our conclusion that K⁺ ions pass through the helix bundle crossing constriction by transient loss and regain of coordinating water molecules is consistent with the experimental data and present knowledge.

Notably, the reviewer offers no alternative interpretation of the data presented in the manuscript; their objection is based purely on an (unreasonable) expectation that unnecessary additional experiments will be carried out (explained below).

At 150 mM K⁺ used in the study, a K⁺ channel typically conducts K⁺ at a rate 10⁶ - 10⁸ M⁻¹ s⁻¹. In order to substantiate the authors' claim, high quality single-channel electric-current recordings (or an alternative, adequate method) from individual channels, which are known to have the constriction, are absolutely necessary.

The flux assay used is not an adequate substitute for the electric-current recordings.

The ion channel field is undergoing change on many fronts and the position that only electrophysiological recordings can provide proof of conduction is no longer a consensus view. A requirement for single channel recordings was not raised by reviewer #1 or #2; admittedly electrophysiology may not be their area of expertise, however they appear satisfied with the experimental proof provided.

In our previous response, we outlined the rationale for using a flux assay. To reiterate, “in single channel recordings in cells or bilayers it would not be possible to state with certainty that measurable currents were not due to a tiny amount (ppm) of non-crosslinked channel in the sample.”

This is a significant issue, given that crosslinking a channel tetramer to completion requires a good deal of optimisation. Even if over 99% completion is achieved, a sample with only one functional channel may be recorded from and therefore false positives are likely. The population-based liposomal ACMA assay we used does not suffer from these shortcomings and unambiguously shows the conduction status of the liposomal population (this is elaborated further below).

Critically, other experts in the field (Mackinnon, in Lee *et al.*, JMB (2009) 387, 1055-1060) have also applied this technique in circumstances where the electrophysiological approach is inappropriate or inadequate (see below).

The K⁺ flux rate of the examined channels estimated from the assay is several orders of magnitude lower than the expected K⁺ permeation rate (Fig. 4b). Regarding the cited data of Glaaser and Schlessinger, the highest flux rate is also extremely low, < 0.01 s⁻¹. It appears that a process, other than K⁺ conduction, in the assay limits the observed rate.

There is no reason to think that this is the case.

We note that several peer reviewed articles, including papers in Nature, Nat Chem Biol, Science, Science Reports, Cell, J Gen Physiol, PNAS, eLife JMB, (some cited in the manuscript) have utilised liposomal ACMA assays to provide evidence of ion channel conduction. In all cases, the rate is ‘several orders of magnitude lower than the expected K⁺ permeation rate’ and is invariably interpreted in a relative, rather than an absolute, way. This is because the basis of the liposomal flux assay is that protonation of the fluorophore, ACMA, causes fluorescence decrease proportional to potassium flux; the rate is thus affected by buffering, ACMA equilibrium, *etc.* Despite this, the signal is sensitive to, and can be titrated with, known regulatory molecules and drug candidates verifying that, regardless of the absolute rates, the activity being assayed is K⁺ conduction.

In 2016 Roderick Mackinnon published an article (Su *et al*, PNAS, 113, 5748-5753, 2016) highlighting ACMA assays as a “novel liposome flux assay that is applicable to most K⁺ channels”. In it, he trialed ACMA assays on the four different classes of K⁺ channel, using the assay to identify small molecule inhibitors by high throughput screening. As in all other cases, the rates of Mackinnon’s liposomal assays were in the same ballpark as ours (and of Glaaser and Schlessinger, and others), i.e. extremely low. However, in each instance the concentration of drug titrated the rate. The IC₅₀ values gained from his ACMA assays on the K⁺ channels were subject to cross-validation by electrophysiological recordings and, despite the non-physiological rates of ACMA assays, the findings of the assays predicted the electrophysiological outcomes.

In his concluding arguments, Rod Mackinnon stated that the assays “**...provide data similar in quality to an electrophysiology assay...**”

We have, therefore, applied the most appropriate approach for the question being addressed and our conclusions are based on, and supported by, the data that was generated from these.

Thus, at best, any data that have thus far been obtained with this method could not address the issue of whether a constriction at the inner helix bundle of Kir channels impedes conduction of K⁺.

The data proves otherwise. In our case, we employed a liposomal ACMA assay to work on a far less complex problem than drug titrations in Su et al. (above). We were essentially looking for a binary result – conducting or not conducting. The assay unambiguously differentiates between the two states and thus our conclusions stand.

Should the apparent kinetics of the flux assay reflect that of K⁺ permeation, the observation of the extremely slow kinetics would strongly reject the notion that the ion conduction was not impeded.

The points made above clarify that this is not the case. The assay distinguishes active from inactive channels, even though the kinetics of ACMA assays are slow in comparison to single channel recordings. Importantly, we observed no significant differences between wild type channels and those with crosslinks that physically constrain the size of the opening at Tyr132. In contrast, the signal from control samples did not significantly differ from empty liposomes, *i.e.* the experimental data and experimental controls are internally consistent.

In our experiments, the classical Kir channel blocker, spermine, blocked conduction from wild type channels as expected, but did not block disulfide-linked channels, revealing that while disulfide-linked channels can conduct K⁺, they do not allow spermine entry, as the disulfide constraints restrict relative movement of the four inner helices. Critically, this shows that K⁺ ions passing Tyr132 must be smaller in cross-section than spermine; this can only be explained by the transient depletion of coordinating water ligands as they pass the constriction.

Reviewers' comments third round:

Reviewer #2 (Remarks to the Author):

Unfortunately, the spermine discrepancy remains unresolved. The authors argue "There is thus no significant discrepancy between our simulation results and the original X-ray crystal structure." and "A significant (~1000-fold) molar excess of spermine over protein in the crystallisation conditions (18) is more than sufficient to account for the ~15 kJ mol⁻¹ energy difference needed to populate position 4, corresponding to the observed X-ray crystallographic position of spermine."

First, in terms of molar excess, the spermine concentrations should be taken into account. In the simulations, 1 spermine per 10.000 water molecules amounts to ~5 mM, whereas in the crystallization conditions, according to ref. 18 a concentration of 50 mM was used. This therefore accounts for a factor of 10 in the bulk concentration, not sufficient to explain the observed difference.

Second, the molar excess argument only applies to difference between bound and unbound states. But the observed discrepancy also applies between the bound states 3 and 4. According to the simulation data, state 3 is about 15 kJ/mol lower in free energy than state 4. Yet state 4 is observed crystallographically.

This discrepancy should be resolved before I can recommend this paper for publication.

Reviewer #2 (Remarks to the Author):

Unfortunately, the spermine discrepancy remains unresolved. The authors argue "There is thus no significant discrepancy between our simulation results and the original X-ray crystal structure." and "A significant (~1000-fold) molar excess of spermine over protein in the crystallisation conditions (18) is more than sufficient to account for the ~15 kJ mol⁻¹ energy difference needed to populate position 4, corresponding to the observed X-ray crystallographic position of spermine."

First, in terms of molar excess, the spermine concentrations should be taken into account. In the simulations, 1 spermine per 10.000 water molecules amounts to ~5 mM, whereas in the crystallization conditions, according to ref. 18 a concentration of 50 mM was used. This therefore accounts for a factor of 10 in the bulk concentration, not sufficient to explain the observed difference.

Second, the molar excess argument only applies to difference between bound and unbound states. But the observed discrepancy also applies between the bound states 3 and 4. According to the simulation data, state 3 is about 15 kJ/mol lower in free energy than state 4. Yet state 4 is observed crystallographically.

This discrepancy should be resolved before I can recommend this paper for publication.

We acknowledge the reviewer's comments that we had not adequately reconciled the differences between our simulations and the X-ray crystal structure. In response, we have examined the crystal structure(s) and carried out additional simulations in order to resolve the discrepancy.

Reviewer #2's point, as we understand it, is that while position ④ reflects the site of spermine in the 2010 structure 2WLK, crystallised at zero field, our MD simulations indicate that, even at zero field, the majority of spermine rests at position ③.

Importantly, the unrestrained MD simulations demonstrate an increase in probability of spermine penetration with field strength (Figure 5d), consistent with published electrophysiological findings (cited) that spermine block is a voltage-dependent process. This indicates that, by and large, the simulations reflect experimental observation.

In the light of reviewer concern, we first revisited the structural data; namely, the electron density maps of 2WLK (originally refined by us). The crystal suffered from largescale systemic disorder (two alternate conformations of the entire pore were refined relative to the cytoplasmic assembly in ~60:40 ratio). Additionally, co-incidence of spermine with a crystal 2-fold (but pseudo 4-fold) axis, exacerbates the positional disorder such that extended spermine density along the conduction pathway is not definitive in terms of the resting position of spermine on the molecular axis (*i.e.* the z-axis). The furthestmost point inside the pore where strong density is observed represents position ④. In 2WLK, spermine is thus modelled at position ④, but an earlier refinement of the same crystal structure (pdb 1XL6, also refined by us) modelled alternate spermine conformations at positions ③ and ④, refining these to a respective occupancy ratio of ~2:1. Neither structure is an exact model, due to the multiple sources of positional disorder, but one cannot rule out that spermine occupies either position ③ or ④ in any particular channel, averaged over all unit cells of the crystal, in accord with our simulations.

In carrying out simulations, we had not previously appreciated that the energetic barrier faced by spermine is dependent on its protonation state. The experimental pKa of spermine is 7.95 (Biophys Chem (1983) 17, 67-74). The microscopic ionisation constants of primary and secondary amines in spermine are nearly identical and indicate comparable amounts of deprotonated primary (N1 or N14) and secondary (N5 or N10) amines at a pH 7.4 (crystallisation condition). The hydrophobic nature of the site where N1 binds in position ④, lined by the sidechains of Leu124 and Ala128, will likely accommodate the deprotonated N1 amine. Bearing this in mind, we conducted additional MD calculations, this time using a spermine deprotonated at N1, to evaluate the effect on the energetic barrier. These new calculations indicated that a spermine with neutral N1 would preferentially rest in position ④, (*i.e.* the energy profile from ③ to ④ is downhill), with the leading neutral amine sitting in the non-polar binding pocket in the cavity.

We submit that the MD data are consistent with spermine occupancy at positions ③ when the spermine is fully charged, or ④ when only triply charged. An ‘N1-deprotonated’ PMF trace is included below. It shows that while the energy is higher at position ④ than ③ for fully protonated spermine, it is lower at position ④ (negative and therefore spontaneous) if the spermine is not protonated at N1.

MD simulations starting with the N1-deprotonated spermine at position ④ were stable in simulations with applied field of 0, 25 and 50 mV/nm (see figure below). In contrast, simulations starting with N1-deprotonated spermine at the ③ position saw spermine rapidly move to position ④. These simulations confirm the relative stability of the N1-deprotonated spermine for position ④ binding.

In this context, the deprotonation and exchange equilibria of spermine are likely to influence the spermine binding equilibria in the crystals. The MD simulations don’t allow proton exchange, and consideration of different protonation states therefore have to be studied independently of one another.

We have revised figure 5 so that panel 5d is now directly comparable to panel c (*i.e.* the positions are now annotated). We have used only the fully charged spermine MD in Figure 5, as it is representative of spermine in solution at neutral pH. The Figure 5 legend has also been revised, to:

“ **Figure 5.** Spermine enters wild type but not disulfide-linked pores. (a) Summary of potassium flux experiments investigating spermine block in wild type and disulfide-linked A133C-T136C channels. Data are shown as mean \pm SEM ($n=3$ or more) experiments; details of replicates are in Extended Data Table 4. Each black or white circle represents the mean of an individual experiment. (b) Comparative dimensions of permeant cations. (c) The PMF represents the energetic barrier to spermine block. Positions 1 to 4 (circles) refer to the positioning of a middle carbon (C8) of spermine relative to the pore – shown in the schematic below. (d) The voltage-dependency of spermine block is illustrated by the trend in probability density of spermine entering the pore at field strengths of 0, 25 and 50 mV nm⁻¹. The shaded region of each panel corresponds to 80% of structures in the MD simulations, shown relative to positions ① to ④. The distance of C8 relative to the centre of mass of Thr96 is annotated for major peaks (red). In the region 18.0 to 27.5 Å, the occupancy sequentially changes from approximately 25 to 40 to 50% with increasing field (0 to 25 to 50 mV nm⁻¹). The field-free distribution of spermine in the cavity predicts only a small probability of spermine occupancy at position ④. ”

In addition, changes to the text were made to three paragraphs on page 6. This now reads

“Steered MD and umbrella sampling simulations were employed to estimate the free energy required to move intracellular spermine into the pore cavity. Figure 5c plots the position of a central methylene carbon, C8, of spermine, as it moves from the cytosol into the cavity, against the PMF. The resultant PMF values indicate a significantly greater energetic barrier to spermine at the Tyr132 collar in disulfide-linked A133C-T136C than in wild type (unconstrained), with respective maxima at site ② of 15 kJ mol⁻¹ for wild type and 25 kJ mol⁻¹ for the disulfide-linked mutant. The 10 kJ mol⁻¹ difference between them infers ~50-fold lower probability of the spermine passing the disulfide-linked constriction of the cysteine pair mutant relative to the unconstrained channel, while 15 kJ mol⁻¹ difference at site ③ approximates a 400-fold lower probability of it reaching that site. The approximate differences in PMF at positions ① to ④ are 5, 10, 15 and 45 kJ mol⁻¹, respectively, with the disulfide-linked mutant always the higher of the two.

In wild type channels at zero field, the energetic barrier of 15 kJ mol⁻¹ faced by spermine as its leading amine N1 passes the Tyr132 collar (Fig. 5c; ①->②) is the same as that experienced as N2 passes the tyrosine collar (③->④). At sites ① and ③, and prior to spermine engaging the channel, the energy is approximately equal. In contrast, in the disulfide-locked pore, there is a significant barrier at site ④ above the PMF at site ③, 40-45 kJ mol⁻¹, preventing spermine from penetrating much beyond the barrier at Tyr132. The MD-predicted differences in free energy of spermine binding at serial sites between wild type and disulfide-linked A133C-T136C KirBac3.1 accord with the experimental assay results that spermine blocks wild type but not disulfide-linked channels.

Probability density profiles corresponding to the position of spermine in the transmembrane cavity at zero field, and 25 and 50 mV nm⁻¹ (Fig. 5d) illustrate the impact of the 15 kJ mol⁻¹ barrier to spermine penetration in wild type channels at different field strengths. At zero field, there is only a small probability that spermine traverses the tyrosine collar and reaches site ③, while at 25 and 50 mV nm⁻¹ progressively more spermine penetrates past the constriction at Tyr132. The calculations indicate that the probability of spermine penetrating further into the pore increases with the field strength applied, in accord with the known voltage-dependence of polyamine block⁴³. Sites ① and ③ exhibit near equivalence in energy under these simulation conditions (noting that ions are absent from the upper cavity).”

With reference to the point that “in terms of molar excess, the spermine concentrations should be taken into account.”, the number of spermine molecules in the simulations should not be used to estimate concentration; the simulations are very artificial in this respect (a larger simulation box would increase the number of water molecules and commensurately decrease the concentration, yet no change would be observed in the simulations).

The reviewer is correct that “the molar excess argument only applies to difference between bound and unbound states”, however, the occupancy of higher energy bound states (i.e. ④ compared to ③) also increases as the concentration of ligand increases. Thus, at low ligand concentrations we would expect to observe only state ③ occupied, and with increasing concentration the occupancy of this state should reach its maximum (1.0) and the occupancy of state ④ should increase. However, the reviewer is correct, the 50 mM spermine is not sufficient to occupy site ④ substantially (see below).

$$P + L \rightarrow P.L_{②} \quad K(a) = [P.L]_{③} / ([P].[L]) \quad \text{equation 1}$$

$$P.L_{③} \rightarrow P.L_{④} \quad K(b) = [P.L]_{④} / [P.L]_{③} \quad \text{equation 2}$$

Substituting P.L_③ from equation 2, [P.L]_③ = [P.L]_④ / K(b), into equation 1,

$$K(a) = [P.L]_{④} / (K(b).[P].[L]) \quad \text{equation 3}$$

And rearranging equation 3 the provide the ratio, Q, of spermine at position ④ compared to ligand free channel

$$Q = [P.L]_{④} / [P] = K(a).K(b).[L]$$

Assuming K(a) ~ 1 (equivalent to ΔG = 0 kJ/mol), K(b) ~ 0.001 (equivalent to ΔG = 15 kJ/mol), and [L] = 50 mM, then Q ~ 5 x 10⁻⁵. Thus, state ④ should not be substantially occupied with spermine fully protonated.

In summary, we present a scenario whereby the preferred position of spermine binding depends on its protonation state. The comparison of crystallographic and MD data has been obfuscated by the uncertainty afforded by the possibility of overlapping positions of spermine and other sources of disorder in the X-ray structures and our lack of appreciation of the ease with which spermine can deprotonate at physiological pH. We submit our MD simulations can be reconciled with the observed crystallographic data and have therefore addressed all the reviewer’s concerns.

Reviewers' comments fourth round:

Reviewer #2 (Remarks to the Author):

the authors have satisfactorily addressed my concerns. I now recommend the paper for publication.